# STABLEVAL: Disagreement-Aware and Stable Evaluation of AI Systems

**Akash Bonagiri** [* 1]   **Gerard Janno Anderias** [* 1]   **Saee Patil** [1]   **Angelina Lai** [1]   **Devang Borkar** [1]   **Gezheng Kang** [1]   **Ishant Gandhi** [1]   **Setareh Rafatirad** [1]   **Houman Homayoun** [1]

## Abstract

Human evaluation remains the primary standard for assessing modern AI systems, yet annotator disagreement, bias, and variability make system rankings fragile under standard majority vote aggregation. Majority vote discards annotator reliability and item-level ambiguity, often yielding unstable comparisons across annotator subsets. We introduce STABLEVAL, a disagreement-aware evaluation framework that models latent item correctness and annotator-specific confusion patterns to produce posterior expected item credit and calibrated agent-level scores. Unlike label-denoising approaches such as Dawid–Skene, STABLEVAL is explicitly designed for stable and uncertainty-aware system evaluation rather than hard label recovery. We formalize ranking stability as a first-class evaluation objective and analyze how aggregation methods preserve or distort underlying annotator behavior. Across controlled synthetic experiments and multiple real-world human-annotated benchmarks, majority vote exhibits increasing score error and ranking instability under annotator heterogeneity and adversarial noise, while STABLEVAL yields more stable and statistically grounded system rankings. These results demonstrate that modeling disagreement is essential for robust and reproducible AI evaluation.

## 1. Introduction

Human evaluation remains the primary standard for assessing modern AI systems (Dutta et al., 2025; Zheng et al., 2023; Bojic et al., 2023). Despite advances in automatic metrics and LLM-based judges, final model comparisons are often grounded in human annotations. However, human judgments are inherently noisy, biased, and heterogeneous (Plank, 2022; Chakraborti et al., 2024). Annotators differ in expertise, strictness, interpretation, and even adversarial behavior, leading to structured disagreement across items.

The dominant aggregation strategy, majority vote treats disagreement as noise to be eliminated. By collapsing multi-annotator judgments into a single hard label, majority vote discards information about annotator reliability and item-level ambiguity (Xu et al., 2024). As a result, system scores and rankings can shift under small changes in annotator composition, raising concerns about robustness and reproducibility in AI evaluation.

We argue that disagreement is not noise to suppress, but signal to model. The central challenge in AI evaluation is not merely recovering a denoised label, but producing stable and uncertainty-aware system comparisons under realistic annotator variability.

To address this challenge, we introduce **STABLEVAL**, a disagreement-aware evaluation framework that models latent item correctness and annotator-specific confusion patterns (Kuzin et al., 2025) to compute posterior expected item credit and calibrated agent-level scores. Unlike classical label-denoising approaches such as Dawid–Skene(Dawid & Skene, 1979), which aim to infer a single latent label per item (Gao et al., 2024), STABLEVAL is explicitly designed for stable system evaluation. Rather than collapsing uncertainty, it preserves graded correctness and structured ambiguity, enabling principled aggregation and confidence-aware ranking.

We further formalize *ranking stability* as a first-class objective for AI system evaluation. Informally, a ranking method is stable if agent orderings remain consistent under subsampling of annotators. We show that majority vote is inherently unstable in the presence of disagreement, whereas STABLEVAL achieves stable rankings in expectation under the assumed generative model.

Across extensive synthetic stress tests and multiple real-world human-annotated benchmarks, we demonstrate that majority vote exhibits increasing score error and ranking volatility under annotator heterogeneity and adversarial noise, while STABLEVAL produces more stable and statistically grounded system comparisons.

---

[*]Equal contribution  [1]Department of Computer Science, University of California, Davis, Davis, CA, USA. Correspondence to: Akash Bonagiri <sbonagiri@ucdavis.edu>.

*Proceedings of the 43rd International Conference on Machine Learning*, Seoul, South Korea. PMLR 306, 2026. Copyright 2026 by the author(s).

**Contributions.** We make four primary contributions:

1. We introduce **STABLEVAL**, a disagreement-aware evaluation framework that produces calibrated posterior item credit instead of hard labels, enabling graded and uncertainty-aware system scoring.

2. We formalize *ranking stability* for AI system evaluation and provide theoretical analysis demonstrating the instability of majority vote under annotator subsampling.

3. We clarify the distinction between *label denoising* and *evaluation stability*, showing that recovering latent labels does not guarantee robust system rankings.

4. We provide extensive empirical validation across synthetic stress tests and multiple human-annotated benchmarks, demonstrating improved stability and robustness under annotator heterogeneity and adversarial noise.

## 2. Related Work

**Disagreement-Aware Annotation Modeling.** There is growing interest in treating annotator disagreement as a meaningful signal rather than noise (Fleisig et al., 2023). Surveys and frameworks highlight the need to model structured disagreement, especially for subjective tasks like toxicity detection and stance annotation (Xu & Jurgens, 2026; Bonagiri et al., 2025; Badam et al., 2022; Akash et al., 2021). Methods that explicitly model annotator-specific behavior or group-level variation via demographic-aware experts and synthetic perspectives demonstrate improved representation of structured disagreement (Xu et al., 2025; Parmar et al., 2023). Query-based multi-annotator behavior modeling also frames disagreement as valuable information rather than noise for consensus and interpretability (Zhang et al., 2025).

**Distributional and Soft-Label Approaches.** Several recent works explore distributional labeling and disagreement modeling for downstream tasks, rather than collapsing to single consensus labels. In semantic textual similarity, modeling full annotation distributions rather than mean scores yields better calibration and ranking accuracy (Wang et al., 2023). Large-scale empirical studies show that aggregating soft labels from crowd annotations improves uncertainty estimation compared to hard majority labels (Wright & Augenstein, 2025). Other work adapts neural architectures to jointly predict item- and annotator-level label distributions, improving soft and perspectivist evaluation metrics (Weerasooriya et al., 2023; Geng, 2024).

**Evaluation and Annotation Variation in Learning.** There is also work examining how disagreements affect modeling and evaluation beyond label aggregation. Studies evaluating whether large language models can capture human annotator disagreements find that models often fail to model disagreement patterns, particularly on tasks that require emotional intelligence or contextual understanding, underscoring the importance of capturing variation in annotations for reliable evaluation (Ni et al., 2026; Calderon et al., 2025; Corso et al., 2023). This aligns with trends noting that conventional majority-based or single-label evaluation overlooks critical aspects of disagreement that influence downstream performance and system comparisons.

**Distinction From Traditional Aggregation.** Compared to classical label-denoising approaches that aim to infer a single ground truth, these recent methods advocate retaining and modeling full disagreement distributions for uncertainty representation, calibration, and richer evaluation (Uma et al., 2021; Fleisig et al., 2023; Beddar-Wiesing et al., 2025). However, few existing approaches formally define or optimize for stable system rankings under annotator variability, which is the gap addressed by STABLEVAL.

## 3. STABLEVAL Framework

We consider the problem of evaluating a set of AI agents using noisy and heterogeneous human judgments. In contrast to classical annotation aggregation settings, where the primary goal is to recover a single consensus or denoised label per item, our objective is to produce stable, uncertainty-aware agent rankings that remain robust under annotator variability and disagreement (Benito-Santos & Ghajari, 2025).

Standard aggregation methods such as majority vote collapse multi-annotator judgments into a single hard label, discarding information about annotator reliability and item-level ambiguity (Uma et al., 2021). STABLEVAL instead models disagreement explicitly and propagates uncertainty through to agent-level scores. Algorithm 1 summarizes the full STABLEVAL procedure, with each component described in detail in the following subsections.

### 3.1. Problem Formulation

Let $\mathcal{A} = \{a_1, \ldots, a_M\}$ denote a set of agents. Each agent produces outputs that are evaluated by multiple human annotators.

An *item $i$* consists of:

- a task prompt,

- an agent-generated response,

- and a set of human annotations.

Each item $i$ is labeled by a subset of annotators $\mathcal{R}_i$. For each item, we assume the existence of an unobserved latent correctness level

$$z_i \in \{0, 1, \ldots, K-1\},$$

representing graded quality (e.g., incorrect, partially correct, correct).

Each annotator $r$ provides an observed label

$$y_{ir} \in \{0, 1, \ldots, K-1\}.$$

Annotators are not assumed to be identical or interchangeable. They may differ systematically in strictness, leniency, expertise, interpretation, or noise characteristics. Our goal is to explicitly model these differences between annotators rather than averaging them away.

### 3.2. Modeling Annotator Behavior

STABLEVAL models each annotator using a confusion matrix

$$\pi_r[c, o] = P(y_{ir} = o \mid z_i = c),$$

which captures the probability that annotator $r$ reports label $o$ when the true correctness level is $c$.

This parameterization allows the model to capture structured behaviors such as consistent strictness (e.g., underreporting high labels), systematic leniency, or random noise. By explicitly modeling annotator-specific confusion patterns, STABLEVAL separates item difficulty from annotator reliability.

We further define class priors

$$\mu_c = P(z_i = c),$$

which represent the overall prevalence of each correctness level in the dataset.

Dirichlet priors are placed over $\mu$ and over each row of $\pi_r$ to enable Bayesian estimation, regularize parameter estimates, and prevent degenerate solutions in low-annotation regimes.

### 3.3. Posterior Inference

We perform Expectation–Maximization (EM) to estimate latent correctness levels and annotator reliability parameters (Algorithm 1, lines 4–11).

In the E-step, we compute the posterior probability of each correctness level:

$$\gamma_{ic} = P(z_i = c \mid \{y_{ir}\}_{r \in \mathcal{R}_i}) = \frac{\mu_c \prod_{r \in \mathcal{R}_i} \pi_r[c, y_{ir}]}{\sum_{c'} \mu_{c'} \prod_{r \in \mathcal{R}_i} \pi_r[c', y_{ir}]}.$$

The quantity $\gamma_{ic}$ represents the model's posterior belief that item $i$ belongs to correctness level $c$, conditioned on all observed annotations and estimated annotator reliabilities.

Importantly, unlike classical label-denoising methods that collapse posterior probabilities into a single hard label (e.g., via $\arg\max_c \gamma_{ic}$), STABLEVAL retains the full posterior distribution. This distinction is central: evaluation stability requires preserving uncertainty rather than discarding it.

### 3.4. Posterior Expected Item Credit

We define the *item credit* as the posterior expected correctness value:

$$\text{credit}(i) = \sum_{c=0}^{K-1} \gamma_{ic} \, v(c),$$

where $v(c)$ is a task-specific scoring function assigning real-valued credit to correctness levels (e.g., $v = [0, 0.5, 1]$).

This formulation generalizes hard-label evaluation. If $\gamma_{ic}$ were degenerate at a single class, the expression reduces to standard deterministic scoring. However, when the annotators disagree or uncertainty is high, partial credit is assigned in proportion to posterior belief. This preserves graded correctness and explicitly encodes item-level ambiguity into downstream evaluation.

### 3.5. Agent Scoring

The score of an agent $a$ is defined as the mean posterior item credit over its evaluated items:

$$S(a) = \frac{1}{|\mathcal{I}_a|} \sum_{i \in \mathcal{I}_a} \text{credit}(i).$$

Because item credit is a posterior expectation, agent scores naturally incorporate annotator reliability and item-level uncertainty.

To quantify uncertainty in agent scores, we use bootstrap resampling over items, yielding confidence intervals for each agent's evaluation score. This enables calibrated comparisons and principled reporting of evaluation variability.

## 3.6. Objective: Stable Evaluation

The objective of STABLEVAL is not latent label recovery per se, but stable and uncertainty-aware agent ranking under annotator variability. By explicitly modeling annotator reliability and propagating posterior uncertainty into item and agent scores, STABLEVAL reduces sensitivity to annotator subsampling and structured disagreement.

This reframing distinguishes STABLEVAL from classical denoising approaches: while both estimate latent correctness, STABLEVAL is designed to support robust system evaluation rather than consensus label extraction.

# 4. Ranking Stability Analysis

A core objective of STABLEVAL is to produce agent rankings that are robust to annotator variability. We therefore formalize *ranking stability* as a first-class evaluation criterion.

## 4.1. Ranking Stability Definition

Let $\mathcal{A}$ denote the set of agents. For a set of annotators $R$, let $S_R : \mathcal{A} \to \mathbb{R}$ denote the scoring function induced by an evaluation method $S$ using annotations from $R$. Let $\mathrm{rank}(S_R)$ denote the ranking of agents induced by $S_R$. Let $\tau_b(\cdot, \cdot)$ denote Kendall's $\tau_b$ rank correlation, which accounts for ties.

Given a full annotator set $R$, we sample a subset $R' \subset R$ uniformly at random without replacement from all subsets of fixed size $|R'| = m$.

We define the ranking stability of method $S$ as

$$\mathrm{Stability}(S) = \mathbb{E}_{R'} \left[ \tau_b(\mathrm{rank}(S_R), \mathrm{rank}(S_{R'})) \right].$$

An evaluation method is perfectly stable if this quantity equals 1, meaning that rankings are invariant under annotator subsampling. This definition captures robustness to annotator composition, which is critical for reproducible AI evaluation.

## 4.2. Instability of Majority Vote

We now characterize the behavior of majority vote under annotator subsampling.

**Proposition 1.** Suppose there exists at least one item whose majority label under $R$ is determined by a single-vote margin. Then, under uniform random subsampling of annotators,

$$\mathbb{E}_{R'} \left[ \tau_b(\mathrm{rank}(S_R^{MV}), \mathrm{rank}(S_{R'}^{MV})) \right] < 1.$$

**Sketch of Proof.** If an item's majority label depends on a one-vote margin, removing the decisive annotator flips its

majority outcome with non-zero probability under random subsampling. Since agent scores are computed as averages over item-level labels, such flips induce score perturbations. When agent score gaps are sufficiently small, these perturbations can alter the induced ranking. Therefore, the probability of a ranking change under subsampling is strictly positive, implying expected stability strictly below one.

This instability arises from the discrete thresholding inherent in majority aggregation.

## 4.3. Asymptotic Stability of STABLEVAL

STABLEVAL replaces discrete majority thresholding with smooth posterior expected credit and explicitly models annotator reliability through confusion matrices.

We analyze stability in the regime where the annotator pool $R$ is fixed and the number of labeled items $N \to \infty$.

**Proposition 2 (Asymptotic Stability of STABLEVAL).** Assume the latent variable model underlying STABLEVAL is correctly specified and identifiable, and that maximum likelihood estimates obtained via EM are consistent as $N \to \infty$.

Let $S_R$ and $S_{R'}$ denote STABLEVAL scores computed using annotator sets $R$ and $R' \subset R$, respectively. Then

$$\max_{a \in \mathcal{A}} \left| S_R(a) - S_{R'}(a) \right| \xrightarrow{p} 0.$$

If limiting agent scores are distinct (i.e., no ties in the population limit), it follows that

$$\mathbb{P}(\mathrm{rank}(S_R) = \mathrm{rank}(S_{R'})) \to 1,$$

equivalently,

$$\mathbb{E}_{R'}[\tau_b(\mathrm{rank}(S_R), \mathrm{rank}(S_{R'}))] \to 1.$$

**Sketch of Argument.** Under correct specification and identifiability, standard consistency results for maximum likelihood estimation in latent variable models imply that parameter estimates converge to their true values as the number of annotated items increases. STABLEVAL scores are smooth functionals of the estimated parameters and average over many items. Consequently, score differences induced by random annotator subsampling vanish asymptotically. When limiting agent scores are separated by a non-zero margin, identical rankings follow with probability tending to one.

Unlike majority vote, which is sensitive to marginal label flips, STABLEVAL produces continuous, reliability-weighted item credit. This smooth aggregation mitigates discrete ranking discontinuities caused by small annotator perturbations.

**Algorithm 1** STABLEVAL: Disagreement-Aware Agent Evaluation

**Require:** Items $\{i\}_{i=1}^N$ with annotations $\{y_{ir}\}_{r \in \mathcal{R}_i}$; agents $\mathcal{A}$ with item sets $\{\mathcal{I}_a\}_{a \in \mathcal{A}}$; correctness levels $K$; credit function $v(\cdot)$; Dirichlet priors $\alpha_\mu, \alpha_\pi$; bootstrap iterations $B$

**Ensure:** Agent scores $\{S(a)\}_{a \in \mathcal{A}}$ with 95% confidence intervals

1: $\hat{z}_i \leftarrow \text{MajorityVote}(\{y_{ir}\}_{r \in \mathcal{R}_i})$ for all $i$
2: $\mu_c \leftarrow \frac{1}{N} \sum_i \mathbf{1}[\hat{z}_i = c]$ for $c \in \{0, \dots, K-1\}$
3: $\pi_r \leftarrow (1-\epsilon) I_K + \frac{\epsilon}{K} \mathbf{1}\mathbf{1}^\top$ for all $r$    (near-identity)
4: **repeat**
5:     **E-step:** for each item $i$, level $c$:
6:         $\gamma_{ic} \leftarrow \dfrac{\mu_c \prod_{r \in \mathcal{R}_i} \pi_r[c, y_{ir}]}{\sum_{c'} \mu_{c'} \prod_{r \in \mathcal{R}_i} \pi_r[c', y_{ir}]}$
7:     **M-step:** update with Dirichlet smoothing:
8:         $\mu_c \leftarrow \dfrac{\sum_i \gamma_{ic} + \alpha_\mu - 1}{\sum_{c'} \sum_i \gamma_{ic'} + K(\alpha_\mu - 1)}$
9:         Let $N_{rco} = \sum_{i: r \in \mathcal{R}_i} \gamma_{ic} \mathbf{1}[y_{ir} = o]$
10:       $\pi_r[c, o] \leftarrow \dfrac{N_{rco} + \alpha_\pi - 1}{\sum_{o'} N_{rco'} + K(\alpha_\pi - 1)}$
11: **until** log-likelihood converges
12: **for** each item $i$ **do**
13:     $\text{credit}(i) \leftarrow \sum_{c=0}^{K-1} \gamma_{ic} v(c)$
14: **end for**
15: **for** each agent $a \in \mathcal{A}$ **do**
16:     $S(a) \leftarrow \dfrac{1}{|\mathcal{I}_a|} \sum_{i \in \mathcal{I}_a} \text{credit}(i)$
17: **end for**
18: **for** $b = 1$ to $B$ **do**
19:     Resample $\mathcal{I}_a$ with replacement; recompute $S^{(b)}(a)$ for each $a$
20: **end for**
21: Compute 95% CI for each $S(a)$ from $\{S^{(b)}(a)\}_{b=1}^B$
22: **return** $\{S(a)\}_{a \in \mathcal{A}}$ with 95% confidence intervals

**STABLEVAL Algorithm**   Algorithm 1 presents the full STABLEVAL procedure, combining EM-based estimation of annotator confusion matrices and class priors with posterior expected credit computation and bootstrap-based uncertainty quantification over agent scores.

# 5. Experiments

We evaluate STABLEVAL across multiple real-world human-annotated benchmarks and controlled synthetic settings. In experiments, STABLEVAL instantiates the model in Section 3 and scores agents using Posterior Expected Credit (PEC). We report it as STABLEVAL/PEC. Our experiments are designed to assess (1) stability under annotator subsampling, (2) robustness to annotator heterogeneity and adversarial noise, and (3) differences in agent-level scores relative to majority vote and Dawid–Skene aggregation.

## 5.1. Datasets

We select four datasets spanning preference evaluation, safety assessment, hate speech detection, and medical summarization. These datasets vary in annotator pool size, agreement rates, and subjectivity, allowing us to test stability under diverse disagreement regimes.

**MT-Bench Human Judgments.**   We use 6,712 annotated examples of expert-level human preference for six models' responses to MT-Bench questions spanning question answering, summarization, and data-to-text tasks (Zheng et al., 2023). Annotations are provided by 65 annotators using a winner-or-tie answering scheme.

To enable unified evaluation, we convert pairwise preference judgments into a three-level correctness scheme: incorrect, partial, and correct. MT-Bench exhibits substantial annotator disagreement, making it a challenging benchmark for stability analysis under heterogeneous annotator behavior.

**ConvAbuse.**   ConvAbuse evaluates two conversational safety agents (E.L.I.Z.A. and CarbonBot) using annotations from eight annotators (Curry et al., 2021). Annotators rate conversations on a five-point abuse severity scale: 1 (not abusive), 0 (ambiguous), -1 (mildly abusive), -2 (strongly abusive), and -3 (very strongly abusive).

We convert this scale to a three-level evaluation scheme where scores of 1 and 0 map to 0 (non-abusive or ambiguous), -1 maps to 1 (mildly abusive), and -2 and -3 map to 2 (strongly or very strongly abusive). Compared to MT-Bench, ConvAbuse exhibits relatively high agreement, allowing us to evaluate whether stability gains persist in low-disagreement regimes.

**QAGS.**   QAGS evaluates summarization factuality for two system-dataset pairs: Bottom-Up Summarization on CNN/DailyMail articles and BART on XSUM articles (Wang et al., 2020). We treat each system-dataset pair as an agent (CNN and XSUM). A total of 169 annotators assess whether each generated summary sentence is factually supported by the source article using a binary label (0 = not supported, 1 = supported).

**MSLR: Multi-Documentation Summarization for Literature Review.**   MSLR evaluates medical evidence summarization using facet-level annotations (population, intervention, outcome, fluency) scored 0–2 (Wang et al., 2022; DeYoung et al., 2021; Wallace et al., 2021). Annotators rate each facet independently. Facet scores are averaged and discretized into aggregate labels: $\geq 1.5 \to$ high quality, $\geq 0.75 \to$ medium quality, and otherwise low quality.

Only (item, agent) pairs with $\geq 2$ independent annotators are retained. Although smaller in size, MSLR provides a

structured, facet-based evaluation setting and enables analysis under limited annotator regimes.

## 5.2. Implementation Details

STABLEVAL is implemented using a Bayesian latent variable model with Dirichlet priors over class priors and annotator confusion matrices. Parameter estimation is performed via Expectation–Maximization.

EM is initialized using majority vote labels and near-identity confusion matrices to encourage stable convergence. We verify convergence empirically by monitoring log-likelihood improvement across iterations.

Agent-level uncertainty is estimated using bootstrap resampling over items (1,000 iterations), yielding 95% confidence intervals for reported scores and stability metrics.

All baselines, including Majority Vote and Dawid–Skene (hard label variant), are implemented using identical preprocessing and evaluation pipelines to ensure fair comparison.

## 5.3. Evaluation Protocol

We evaluate methods along four dimensions:

- **Agent Scores:** Mean agent-level credit under each aggregation method.
- **Score Adjustments:** Differences between STABLEVAL scores and majority vote.
- **Annotator Diagnostics:** Estimated annotator accuracy, leniency, and strictness derived from confusion matrices.
- **Item Ambiguity:** Entropy of posterior correctness distributions.
- **Ranking Stability:** Kendall's Tau correlation under random annotator subsampling.

Ranking stability is evaluated by repeatedly sampling subsets of annotators and measuring correlation between rankings produced using the full and subsampled annotator sets. This directly tests robustness to annotator composition, a core objective of STABLEVAL.

## 5.4. Computational Resources

The ranking stability analysis constitutes the most computationally demanding component, as it requires executing the full annotation-aggregation-evaluation pipeline across $R = 10$ repetitions per experimental configuration to compute pairwise Kendall's $\tau$ rank correlations. To mitigate runtime, we parallelized computations across CPU cores using Python's `multiprocessing` module.

All experiments were conducted on a single machine equipped with an Intel Core i9 processor. The complete ablation study was completed in approximately 5 hours. All

remaining experiments incurred negligible runtime overhead.

## 6. Results

### 6.1. Synthetic Stress Tests

We evaluate aggregation methods across six synthetic configurations (Figure 1) designed to isolate annotator heterogeneity, adversarial behavior, item difficulty, agent quality gaps, and annotation density (Table 4).

#### 6.1.1. LABEL RECOVERY VS. EVALUATION STABILITY.

Across all synthetic settings, Dawid–Skene achieves the lowest MSE with respect to latent ground truth (Table 4), reflecting its objective of label denoising. Posterior Expected Credit (PEC) consistently outperforms Majority Vote in MSE, though it remains slightly above Dawid–Skene. Importantly, low MSE does not necessarily imply stable agent rankings. We therefore analyze ranking behavior and stability under annotator subsampling.

#### 6.1.2. ADVERSARIAL AND HETEROGENEOUS ANNOTATORS.

Under adversarial conditions (up to 40% adversarial annotators), performance gaps widen substantially (Figure 3). While Dawid–Skene maintains the lowest MSE, Majority Vote degrades sharply (Table 4). Despite this, all methods maintain ranking accuracy above 98.8%, indicating that coarse ranking may remain preserved even under substantial noise (Table 6).

Under heterogeneous strict and lenient annotator populations, Majority Vote shows increasing MSE and instability, while both Dawid–Skene and PEC remain comparatively robust (Figure 4). PEC consistently improves over Majority Vote in both error and ranking consistency.

#### 6.1.3. AGENT QUALITY GAPS AND HARD ITEMS.

When agent quality gaps are tight, ranking becomes more sensitive to noise (Figure 9). In this regime, PEC and Dawid–Skene achieve higher Kendall's Tau correlations than Majority Vote (Table 7). Hard items increase error for all methods (Figure 10); however, discrete majority thresholding causes sharper ranking sensitivity. PEC achieves perfect Kendall's Tau (1.000) in the no-hard-item regime (Figure 11), illustrating the benefit of smooth posterior aggregation.

**Annotation Density.** Increasing labels per item reduces MSE for all methods (Figure 12). While Dawid–Skene maintains the lowest MSE across annotation densities, PEC

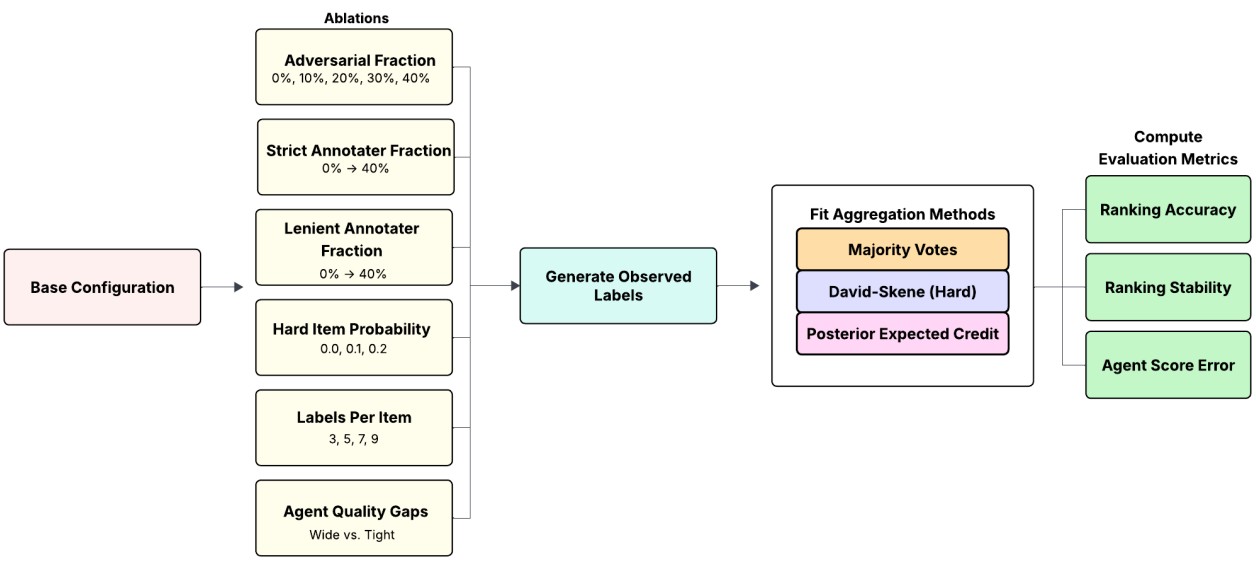

*Figure 1.* Synthetic Evaluation Pipeline. Starting from a base configuration, we systematically vary six ablation parameters: adversarial fraction, strict and lenient annotator fractions, hard item probability, labels per item, and agent quality gaps. For each configuration, we generate observed labels, fit three aggregation methods Majority Vote, Dawid–Skene (Hard), and Posterior Expected Credit and compute evaluation metrics: ranking accuracy, ranking stability, and agent score error.

consistently outperforms Majority Vote and achieves near-perfect ranking stability at moderate label counts (Table 5).

These results highlight a key distinction: Dawid–Skene optimizes label recovery, while STABLEVAL (via PEC) prioritizes smooth and stability-preserving aggregation. In high-density regimes, all methods converge; differences are most pronounced in realistic low-to-moderate annotation settings.

### 6.2. Real-World Benchmarks

#### 6.2.1. MT-BENCH

MT-Bench exhibits substantial annotator disagreement (16.6% unanimous agreement; 53% average pairwise agreement), making it a stress test for stability under heterogeneous judgments (Table 1).

While agent-level mean scores differ across aggregation methods (Figure 14), ranking stability analysis shows PEC produces the most stable rankings under annotator subsampling, with Dawid–Skene intermediate and Majority Vote least stable (Table 9).

A modest but meaningful fraction of responses (8.23%) exhibit high ambiguity, suggesting genuine label uncertainty beyond annotator noise (Table 12). Annotator diagnostics reveal substantial variation in estimated reliability (Table 10), which directly influences score adjustments under STABLEVAL.

#### 6.2.2. CONVABUSE

ConvAbuse exhibits high agreement (78.4% unanimous; 84.4% pairwise), providing a low-disagreement regime (Table 2). In this setting, all methods produce identical agent rankings and perfect stability (Table 14).

*Table 1.* MT-Bench dataset summary statistics.

| Metric | Value |
|---|---|
| Total annotations | 6,710 |
| Unique responses | 960 |
| Unique agents | 6 |
| Unique annotators | 65 |
| Avg. annotations per response | 6.99 |
| Unanimous agreement | 16.6% |
| Pairwise agreement | 53.0% |

*Table 2.* ConvAbuse dataset summary statistics.

| Metric | Value |
|---|---|
| Total annotations | 9,571 |
| Unique responses | 2,894 |
| Unique agents | 2 |
| Unique annotators | 8 |
| Avg. annotations per response | 3.31 |
| Unanimous agreement | 78.4% |
| Pairwise agreement | 84.4% |

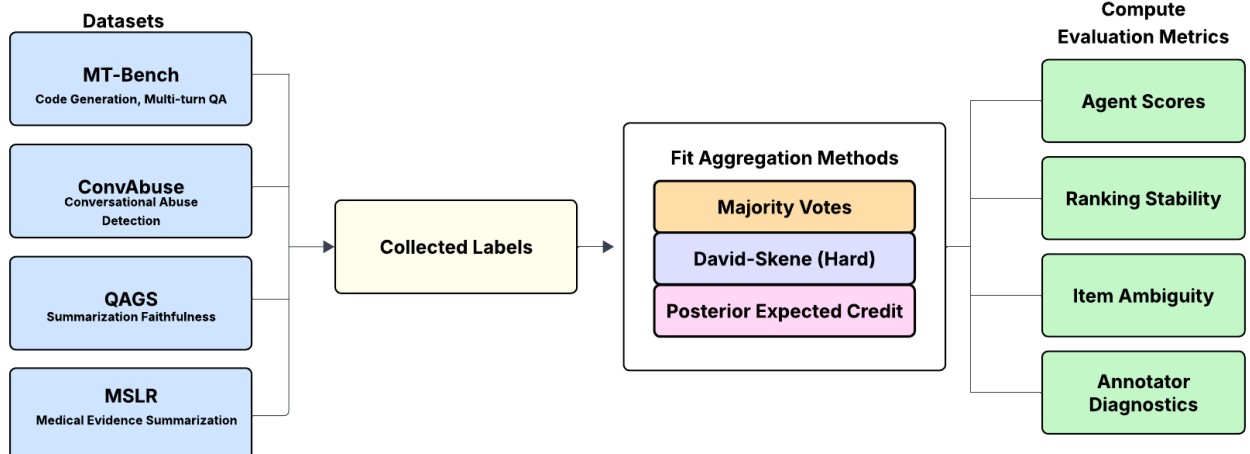

*Figure 2.* Real dataset evaluation pipeline. Four benchmark datasets (MT-Bench, ConvAbuse, QAGS, MSLR) with collected human labels are aggregated using three methods: Majority Votes, Dawid–Skene (Hard), and Posterior Expected Credit. Agent scores are computed and evaluated across four metrics: Agent Scores, Ranking Stability, Item Ambiguity, and Annotator Diagnostics.

Score differences between methods are small (Table 13), demonstrating that STABLEVAL does not introduce instability in high-agreement regimes. Annotator diagnostics highlight measurable reliability differences (Table 15), but these do not alter ranking outcomes due to strong label consensus.

### 6.2.3. QAGS

This dataset features a large annotator pool (169 annotators) with moderate disagreement (Table 18). All methods achieve perfect stability in this dataset (Table 20).

The top-10 annotators show near-perfect accuracy estimates (Table 21), suggesting that disagreement is limited and reliability modeling provides marginal ranking adjustments. In such high-consensus regimes, aggregation method choice has minimal impact on stability.

### 6.2.4. MSLR

MSLR represents a small-annotator, moderate-disagreement regime (Table 24). Under this constrained setting, ranking variability increases for all methods (Table 25).

Majority Vote and Dawid–Skene exhibit slightly lower variability than PEC (Table 26), reflecting the limited sample size and reduced reliability signal. However, PEC provides richer item-level ambiguity estimates (Table 28), which may be beneficial for downstream uncertainty reporting.

### 6.3. Summary of Empirical Findings

Across synthetic and real-world benchmarks, we observe:

- Dawid–Skene achieves the lowest latent-label MSE, consistent with its denoising objective.

- Posterior Expected Credit consistently improves over Majority Vote in error and ranking stability under annotator heterogeneity.

- Stability gains are most pronounced in high-disagreement regimes (e.g., MT-Bench).

- In high-consensus settings, all methods converge to similar rankings.

Gains are largest in high-disagreement regimes; in high-consensus regimes, MV can already be perfectly stable, and DS/PEC may add mild variability. These findings support the core claim of STABLEVAL: explicitly modeling disagreement and preserving posterior uncertainty improves robustness and stability in realistic evaluation settings, particularly when annotator variability is substantial.

## 7. Discussion

### 7.1. When Does Disagreement-Aware Evaluation Matter?

Our results suggest that modeling annotator disagreement is most impactful in evaluation settings characterized by heterogeneous judgments, ambiguous items, or limited annotator consensus. In high-disagreement regimes such as MT-Bench, where unanimous agreement is rare, and annotator

perspectives vary widely, STABLEVAL produces substantially more stable agent rankings than majority vote and classical aggregation methods. In contrast, in high-consensus settings such as ConvAbuse, all aggregation methods converge to similar rankings, and disagreement-aware modeling provides limited additional benefit. This indicates that the value of STABLEVAL lies not in universally outperforming simpler methods, but in improving robustness precisely when disagreement is informative rather than incidental.

## 7.2. Evaluation Stability vs. Label Recovery.

A central theme of this work is the distinction between latent label recovery and stable system evaluation. While Dawid–Skene consistently achieves lower error with respect to latent ground truth in synthetic settings, this does not necessarily translate into more stable agent rankings under annotator subsampling. STABLEVAL is explicitly designed to preserve uncertainty and mitigate ranking volatility caused by marginal annotation changes, highlighting that stability is a distinct and practically relevant evaluation objective.

## 7.3. Implications for Human and Model-Based Evaluation.

As AI systems are increasingly evaluated using human judgments or LLM-based judges, annotator variability and disagreement are unavoidable. Our findings suggest that evaluation pipelines should model uncertainty and annotator behavior rather than collapsing judgments into hard labels. STABLEVAL provides a principled framework for doing so, and can be applied to both human annotation settings and model-based evaluation scenarios where multiple judges exhibit systematic biases or calibration differences.

## 7.4. Interpretability and Diagnostic Value.

Beyond producing agent scores, STABLEVAL yields interpretable diagnostics at both the annotator and item level. Estimated annotator confusion matrices reveal patterns of strictness, leniency, and reliability, while posterior item ambiguity highlights evaluation instances where judgments are intrinsically uncertain. These diagnostics can inform annotation quality control, benchmark refinement, and targeted data collection strategies.

## 7.5. Limitations

STABLEVAL prioritizes ranking stability under annotator variability rather than exact latent label recovery; consequently, denoising-oriented methods such as Dawid–Skene may achieve lower mean squared error in synthetic settings where ground truth is known. Empirical gains are most pronounced in high-disagreement regimes with heterogeneous annotator behavior, while in high-consensus settings

majority vote and reliability-aware approaches often yield similar rankings. The framework assumes conditional independence of annotators given latent correctness and models reliability through confusion matrices, which may not hold under correlated biases, shared training artifacts, or social influence. Additionally, STABLEVAL introduces computational overhead and requires sufficient annotation density to estimate annotator reliability; in sparse-label or very small-annotator regimes, estimates may be unstable. Finally, to enable unified comparison across benchmarks, we discretize continuous or multi-level labels into a three-level correctness scheme; alternative discretizations or task-specific credit mappings could affect quantitative outcomes.

## 7.6. Future Directions.

Future work could extend STABLEVAL to model annotator dependencies, task-specific difficulty, or hierarchical correctness structures. Integrating disagreement-aware evaluation with adaptive annotation strategies or uncertainty-aware benchmarking protocols also represents a promising direction for improving the reliability of AI evaluation.

## 8. Conclusion

We introduced STABLEVAL, a disagreement-aware framework for stable and uncertainty-aware evaluation of AI systems. Rather than collapsing multi-annotator judgments into hard consensus labels, STABLEVAL models annotator-specific reliability and preserves posterior uncertainty through posterior expected item credit. This design enables principled aggregation of heterogeneous judgments and produces agent rankings that are more robust to annotator subsampling. Across synthetic stress tests and diverse real-world benchmarks, we observed a consistent pattern: while classical denoising methods such as Dawid–Skene achieve lower latent-label error, stability under annotator variability is a distinct objective. In high-disagreement regimes, STABLEVAL yields more stable rankings and mitigates volatility introduced by marginal annotation changes. In high-consensus settings, aggregation methods converge, indicating that disagreement-aware modeling does not introduce instability when disagreement is minimal. These findings suggest that evaluation pipelines should move beyond majority vote and explicitly model annotator behavior when disagreement is substantial. As human and model-based evaluation increasingly guide model selection, safety assessment, and benchmark comparisons, robust and reproducible aggregation becomes critical. STABLEVAL provides a principled foundation for disagreement-aware evaluation, reframing stability as a first-class objective and highlighting the importance of modeling uncertainty in modern AI benchmarking.

## Impact Statement

This work proposes STABLEVAL, a framework for disagreement-aware and stability-oriented evaluation of AI systems. By explicitly modeling annotator reliability and preserving uncertainty in multi-annotator settings, STABLEVAL aims to improve the robustness and reproducibility of system comparisons. More stable evaluation pipelines can reduce the risk of selecting models based on fragile or annotator-dependent rankings, which is particularly important in high-stakes domains such as safety assessment, content moderation, and medical or legal benchmarking.

A positive impact of this work is promoting transparency in evaluation by highlighting annotator heterogeneity and item-level ambiguity rather than collapsing disagreement into a single consensus label. This may encourage more careful interpretation of benchmark results and reduce overconfidence in marginal performance differences.

However, disagreement-aware aggregation also introduces modeling assumptions about annotator behavior. If misapplied in settings with insufficient data or inappropriate assumptions, such models could obscure genuine minority perspectives or overcorrect for annotator variability. Additionally, more stable rankings do not necessarily imply fair or unbiased evaluation; biases present in the annotation process may still propagate through reliability-aware modeling.

Overall, this work contributes toward more statistically grounded and transparent evaluation practices, while emphasizing the importance of understanding the limits and assumptions underlying aggregation methods.

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

# Appendix

## A. Dataset and Code Access

All code used for dataset processing and evaluation is available at:

- Code & Data:
  https://github.com/BSAkash/STABLEVAL

## B. Dataset Licenses

All datasets used in this study are publicly available and licensed for academic use. The following licenses apply to each source:

**lmsys/mt_bench_human_judgments** - All materials are licensed under the Creative Commons Attribution-NonCommercial-ShareAlike 4.0 International License

**ConvAbuse: Data, Analysis, and Benchmarks for Nuanced Abuse Detection in Conversational AI** - All materials are licensed under the Creative Commons Attribution-NonCommercial-ShareAlike 4.0 International License.

**allenai/mslr-shared-task** - All materials are licensed under the Apache License, Version 2.0.

**qags: Question Answering and Generation for Summarization** - All materials are licensed under the Creative Commons Attribution-NonCommercial-ShareAlike 4.0 International License

## C. MT-Bench Conversion Sensitivity

MT-Bench is originally a three-outcome pairwise preference task (win/tie/loss). Our conversion preserves this ordinal structure while reinterpreting it within a latent correctness framework, without introducing additional thresholds or altering relative comparisons. While MT-Bench provides relative preference judgments, STABLEVAL requires a per-response correctness formulation. Our mapping is a monotonic transformation that preserves relative ordering, ensuring that disagreement structure is retained.

To directly address sensitivity to the MT-Bench conversion, we evaluated multiple alternative mapping schemes, including binary formulations (dropping ties, merging ties with wins or losses) and three-class formulations with different tie weights. As shown in Table 3, while absolute scores vary across mappings, the relative ranking of agents remains stable across all configurations. The sole rank exchange occurs between gpt-4 and gpt-3.5-turbo under the Tie→Win mapping, where ties are awarded as wins and the two top systems are statistically indistinguishable; ranks 3 through 6 are invariant across all six mappings. This demonstrates that our results are not driven by a specific mapping choice,

but are robust to how pairwise preferences are converted, and that the observed gains arise from modeling annotator disagreement rather than the conversion itself.

*Table 3.* MT-Bench conversion sensitivity. PEC agent scores under alternative pairwise-preference-to-correctness mappings. Ranks shown in parentheses. Baseline corresponds to the win/tie/loss → correct/partial/incorrect mapping used in the main results.

| Agent | Baseline | DropTies | Tie→Win | Tie→Loss | Tie=0.25 | Tie=0.75 |
|---|---|---|---|---|---|---|
| gpt-4 | 0.847 (1) | 0.873 (1) | 0.930 (2) | 0.754 (1) | 0.806 (1) | 0.888 (1) |
| gpt-3.5-turbo | 0.840 (2) | 0.870 (2) | 0.939 (1) | 0.724 (2) | 0.792 (2) | 0.887 (2) |
| claude-v1 | 0.778 (3) | 0.795 (3) | 0.869 (3) | 0.668 (3) | 0.730 (3) | 0.826 (3) |
| vicuna-13b-v1.2 | 0.622 (4) | 0.579 (4) | 0.706 (4) | 0.441 (4) | 0.564 (4) | 0.681 (4) |
| alpaca-13b | 0.290 (5) | 0.222 (5) | 0.406 (5) | 0.154 (5) | 0.235 (5) | 0.344 (5) |
| llama-13b | 0.131 (6) | 0.090 (6) | 0.222 (6) | 0.061 (6) | 0.096 (6) | 0.165 (6) |

# D. Additional Evaluation Metrics

The following tables present additional evaluation metrics across synthetic configurations: Mean Squared Error comparison, ranking accuracy, Kendall's Tau rank correlation summary, and stability across varying label counts.

*Table 4.* Summary of MSE ($\times 10^{-3}$) across synthetic configurations. Best results in **bold**.

| Configuration | Setting | MV | DS | PEC |
|---|---|---|---|---|
| Labels per Item | 3 labels | 4.78 | **3.59** | 5.71 |
| | 5 labels | 2.62 | **1.56** | 2.36 |
| | 7 labels | 1.73 | **1.02** | 1.44 |
| | 9 labels | 1.09 | **0.70** | 0.96 |
| Agent Quality Gap | Tight | 0.59 | **0.46** | 0.72 |
| | Wide | 2.62 | **1.56** | 2.36 |
| Adversarial Annotators | 0% | 2.02 | **1.38** | 2.05 |
| | 10% | 3.14 | **1.69** | 2.58 |
| | 20% | 4.29 | **2.04** | 3.21 |
| | 30% | 6.31 | **2.57** | 4.11 |
| | 40% | 8.42 | **3.47** | 5.54 |
| Strict Annotators | 0% | 1.93 | **1.31** | 1.79 |
| | 10% | 1.85 | **1.32** | 1.90 |
| | 20% | 2.02 | **1.38** | 2.05 |
| | 30% | 2.53 | **1.51** | 2.29 |
| | 40% | 3.65 | **1.69** | 2.60 |
| Lenient Annotators | 0% | 1.77 | **1.23** | 1.77 |
| | 10% | 1.89 | **1.33** | 1.97 |
| | 20% | 2.33 | **1.51** | 2.23 |
| | 30% | 3.14 | **1.65** | 2.51 |
| | 40% | 4.44 | **1.87** | 2.86 |
| Hard Items | 0% | 1.10 | **0.36** | 0.60 |
| | 10% | 1.78 | **0.83** | 1.29 |
| | 20% | 2.62 | **1.56** | 2.36 |

*Table 5.* Summary of Stability scores across synthetic configurations. Higher values indicate more stable rankings. Best results in **bold**.

| Configuration | Setting | MV | DS | PEC |
|---|---|---|---|---|
| Labels per Item | 3 labels | 0.985 | **0.989** | **0.989** |
| | 5 labels | 0.988 | **0.991** | **0.991** |
| | 7 labels | 0.987 | **0.992** | **0.992** |
| | 9 labels | 0.987 | **0.992** | **0.992** |

*Table 6.* Summary of Ranking Accuracy (%) across synthetic configurations. Best results in **bold**.

| Configuration | Setting | MV | DS | PEC |
|---|---|---|---|---|
| Labels per Item | 3 labels | 99.2 | 99.2 | **99.6** |
| | 5 labels | 99.8 | **99.9** | **99.9** |
| | 7 labels | 99.8 | 99.7 | **99.7** |
| | 9 labels | **100.0** | 99.8 | 99.9 |
| Agent Quality Gap | Tight | 96.9 | 97.6 | **97.6** |
| | Wide | 99.8 | **99.9** | **99.9** |
| Adversarial Annotators | 0% | 99.8 | 99.8 | **99.9** |
| | 10% | **99.9** | **99.9** | **99.9** |
| | 20% | 99.6 | 99.5 | **99.7** |
| | 30% | 99.2 | 99.6 | **99.8** |
| | 40% | 98.8 | 99.7 | **99.8** |
| Strict Annotators | 0% | **99.9** | 99.8 | **99.9** |
| | 10% | **99.9** | 99.6 | **99.9** |
| | 20% | 99.8 | 99.8 | **99.9** |
| | 30% | 99.7 | **99.9** | **99.9** |
| | 40% | 99.7 | 99.8 | **99.8** |
| Lenient Annotators | 0% | 99.8 | 99.8 | **99.9** |
| | 10% | **99.9** | 99.8 | **99.9** |
| | 20% | 99.8 | **99.9** | **99.9** |
| | 30% | 99.7 | 99.8 | **99.7** |
| | 40% | 99.5 | **99.7** | **99.7** |
| Hard Items | 0% | 99.9 | **100.0** | **100.0** |
| | 10% | 99.6 | 99.8 | **99.9** |
| | 20% | 99.8 | **99.9** | **99.9** |

*Table 7.* Summary of Kendall's Tau correlation across synthetic configurations. Best results in **bold**.

| Configuration | Setting | MV | DS | PEC |
|---|---|---|---|---|
| Labels per Item | 3 labels | 0.985 | 0.984 | **0.992** |
| | 5 labels | 0.996 | **0.997** | **0.997** |
| | 7 labels | 0.996 | 0.993 | **0.995** |
| | 9 labels | **1.000** | 0.997 | 0.997 |
| Agent Quality Gap | Tight | 0.939 | **0.952** | **0.953** |
| | Wide | 0.996 | **0.997** | **0.997** |
| Adversarial Annotators | 0% | 0.996 | 0.997 | **0.997** |
| | 10% | 0.997 | 0.997 | **0.999** |
| | 20% | 0.993 | 0.991 | **0.995** |
| | 30% | 0.984 | 0.992 | **0.996** |
| | 40% | 0.977 | 0.993 | **0.996** |
| Strict Annotators | 0% | **0.997** | 0.997 | **0.997** |
| | 10% | **0.997** | 0.992 | **0.997** |
| | 20% | 0.996 | 0.997 | **0.997** |
| | 30% | 0.993 | **0.997** | **0.997** |
| | 40% | 0.994 | 0.996 | **0.996** |
| Lenient Annotators | 0% | 0.996 | 0.996 | **0.997** |
| | 10% | **0.997** | 0.996 | **0.997** |
| | 20% | 0.997 | **0.997** | **0.997** |
| | 30% | 0.993 | **0.996** | 0.995 |
| | 40% | 0.991 | **0.995** | **0.995** |
| Hard Items | 0% | 0.997 | **1.000** | **1.000** |
| | 10% | 0.992 | 0.996 | **0.997** |
| | 20% | 0.996 | **0.997** | **0.997** |

The following tables present detailed results for MT-Bench, including dataset statistics, agent scores across aggregation methods, ranking stability, and annotator quality.

*Table 8.* Agent scores on MT-Bench across scoring methods.

| Agent | MV | DS | PEC | Δ(PEC-MV) | Items |
|---|---|---|---|---|---|
| gpt-4 | 0.791 | 0.856 | 0.847 | +0.056 | 160 |
| gpt-3.5-turbo | 0.781 | 0.834 | 0.840 | +0.059 | 160 |
| claude-v1 | 0.744 | 0.794 | 0.778 | +0.035 | 160 |
| vicuna-13b-v1.2 | 0.478 | 0.619 | 0.622 | +0.144 | 160 |
| alpaca-13b | 0.150 | 0.284 | 0.290 | +0.140 | 160 |
| llama-13b | 0.084 | 0.113 | 0.131 | +0.046 | 160 |

*Table 9.* Ranking stability on MT-Bench (lower is more stable).

| Method | Mean Rank Std | Mean Rank Range |
|---|---|---|
| Majority Vote | 0.259 | 1.000 |
| Dawid–Skene (Hard) | 0.223 | 0.667 |
| Posterior Expected Credit | 0.197 | 0.667 |

*Table 10.* Top 10 annotators by learned accuracy on MT-Bench.

| Annotator | Accuracy | Leniency | Strictness |
|---|---|---|---|
| expert_3 | 0.821 | 0.296 | 0.356 |
| expert_53 | 0.806 | 0.281 | 0.357 |
| expert_55 | 0.778 | 0.276 | 0.388 |
| expert_46 | 0.771 | 0.239 | 0.352 |
| expert_26 | 0.754 | 0.336 | 0.337 |
| author_2 | 0.753 | 0.286 | 0.308 |
| expert_2 | 0.748 | 0.275 | 0.355 |
| expert_27 | 0.746 | 0.311 | 0.386 |
| expert_40 | 0.742 | 0.316 | 0.366 |
| expert_9 | 0.742 | 0.216 | 0.354 |

*Table 11.* Most ambiguous responses on MT-Bench.

| Response ID | Ambiguity | Confidence | Pred. |
|---|---|---|---|
| mtbench_87_turn2_claude-v1 | 0.597 | 0.403 | 1 |
| mtbench_125_turn1_claude-v1 | 0.565 | 0.435 | 2 |
| mtbench_101_turn2_alpaca-13b | 0.551 | 0.449 | 2 |
| mtbench_86_turn1_llama-13b | 0.550 | 0.450 | 1 |
| mtbench_95_turn1_alpaca-13b | 0.543 | 0.457 | 1 |
| mtbench_90_turn1_claude-v1 | 0.539 | 0.461 | 1 |
| mtbench_117_turn1_claude-v1 | 0.513 | 0.487 | 2 |
| mtbench_86_turn1_claude-v1 | 0.505 | 0.495 | 1 |
| mtbench_130_turn2_alpaca-13b | 0.499 | 0.501 | 1 |
| mtbench_139_turn1_alpaca-13b | 0.497 | 0.503 | 0 |

*Table 12.* Key findings on MT-Bench.

| Metric | Value |
|---|---|
| Score correlation (MV vs PEC) | 0.989 |
| Maximum score change | 0.144 |
| Agents with rank changes | 0 / 6 |
| Average item ambiguity | 0.065 |
| Highly ambiguous items ($>0.3$) | 79 (8.23%) |

The following tables present evaluation metrics for ConvAbuse including dataset summary statistics, agent score comparison across aggregation methods, ranking stability analysis, and annotator quality.

*Table 13.* Agent scores on ConvAbuse across scoring methods.

| Agent | MV | DS | PEC | Δ(PEC-MV) | Items |
|---|---|---|---|---|---|
| E.L.I.Z.A. | 0.154 | 0.186 | 0.193 | +0.039 | 2547 |
| CarbonBot | 0.029 | 0.039 | 0.043 | +0.014 | 347 |

*Table 14.* Ranking stability on ConvAbuse (lower is more stable).

| Method | Mean Rank Std | Mean Rank Range |
|---|---|---|
| Majority Vote | 0.000 | 0.000 |
| Dawid–Skene (Hard) | 0.000 | 0.000 |
| Posterior Expected Credit | 0.000 | 0.000 |

*Table 15.* Annotator quality on ConvAbuse (all 8 annotators).

| Annotator | Accuracy | Leniency | Strictness |
|---|---|---|---|
| Annotator1 | 0.799 | 0.363 | 0.399 |
| Annotator4 | 0.798 | 0.374 | 0.463 |
| Annotator5 | 0.782 | 0.477 | 0.329 |
| Annotator2 | 0.760 | 0.338 | 0.473 |
| Annotator6 | 0.751 | 0.347 | 0.506 |
| Annotator8 | 0.733 | 0.364 | 0.499 |
| Annotator3 | 0.647 | 0.258 | 0.567 |
| Annotator7 | 0.519 | 0.181 | 0.745 |

*Table 16.* Most ambiguous responses on ConvAbuse.

| Response ID | Ambiguity | Confidence | Pred. |
|---|---|---|---|
| convabuse_298297.0_E.L.I.Z.A. | 0.564 | 0.436 | 1 |
| convabuse_193305.0_E.L.I.Z.A. | 0.497 | 0.503 | 1 |
| convabuse_104335.0_E.L.I.Z.A. | 0.491 | 0.509 | 1 |
| convabuse_261585.0_E.L.I.Z.A. | 0.488 | 0.512 | 1 |
| convabuse_321553.0_E.L.I.Z.A. | 0.486 | 0.514 | 2 |
| convabuse_301675.0_E.L.I.Z.A. | 0.483 | 0.517 | 1 |
| convabuse_244402.0_E.L.I.Z.A. | 0.471 | 0.529 | 1 |
| convabuse_444222.0_E.L.I.Z.A. | 0.469 | 0.531 | 1 |
| convabuse_92200.0_E.L.I.Z.A. | 0.468 | 0.532 | 1 |
| convabuse_24975.0_E.L.I.Z.A. | 0.462 | 0.538 | 1 |

*Table 17.* Key findings on ConvAbuse.

| Metric | Value |
|---|---|
| Score correlation (MV vs PEC) | 1.000 |
| Maximum score change | 0.039 |
| Agents with rank changes | 0 / 2 |
| Average item ambiguity | 0.049 |
| Highly ambiguous items ($>0.3$) | 107 (3.70%) |

The following tables present evaluation metrics for QAGS, a summarization factuality dataset with 2 models (CNN, XSUM) rated by 169 crowdworkers: dataset summary statistics, agent score comparison across aggregation methods, ranking stability analysis, and annotator quality.

*Table 18.* QAGS dataset summary statistics.

| Metric | Value |
|---|---|
| Total annotations | 2,859 |
| Unique responses | 953 |
| Unique agents | 2 |
| Unique annotators | 169 |
| Unanimous agreement | 65.6% |
| Pairwise agreement | 77.1% |

*Table 19.* Agent scores on QAGS across scoring methods.

| Agent | MV | DS | PEC | $\Delta$(PEC-MV) | Items |
|---|---|---|---|---|---|
| CNN | 0.744 | 0.723 | 0.722 | $-0.022$ | 714 |
| XSUM | 0.485 | 0.531 | 0.529 | $+0.044$ | 239 |

*Table 20.* Ranking stability on QAGS (lower is more stable).

| Method | Mean Rank Std | Mean Rank Range |
|---|---|---|
| Majority Vote | 0.000 | 0.000 |
| Dawid–Skene (Hard) | 0.000 | 0.000 |
| Posterior Expected Credit | 0.000 | 0.000 |

*Table 21.* Top 10 annotators by accuracy on QAGS.

| Annotator | Accuracy | Leniency | Strictness |
|---|---|---|---|
| worker_171 | 0.999 | 0.500 | 0.500 |
| worker_13 | 0.998 | 0.501 | 0.499 |
| worker_85 | 0.998 | 0.500 | 0.500 |
| worker_160 | 0.998 | 0.501 | 0.499 |
| worker_155 | 0.998 | 0.501 | 0.499 |
| worker_82 | 0.997 | 0.499 | 0.501 |
| worker_158 | 0.997 | 0.500 | 0.500 |
| worker_54 | 0.997 | 0.501 | 0.499 |
| worker_73 | 0.997 | 0.500 | 0.500 |
| worker_74 | 0.996 | 0.501 | 0.499 |

*Table 22.* Most ambiguous responses on QAGS.

| Response ID | Amb. | Conf. | Pred. |
|---|---|---|---|
| cnn_227_sent2_CNN | 0.500 | 0.500 | 1 |
| xsum_171_sent0_XSUM | 0.496 | 0.504 | 1 |
| cnn_143_sent0_CNN | 0.477 | 0.523 | 1 |
| cnn_21_sent1_CNN | 0.477 | 0.523 | 1 |
| cnn_65_sent2_CNN | 0.473 | 0.527 | 0 |
| cnn_81_sent2_CNN | 0.470 | 0.530 | 0 |
| cnn_111_sent0_CNN | 0.466 | 0.534 | 0 |
| cnn_87_sent1_CNN | 0.465 | 0.535 | 0 |
| xsum_28_sent0_XSUM | 0.462 | 0.538 | 0 |
| cnn_115_sent0_CNN | 0.460 | 0.540 | 0 |

*Table 23.* Key findings on QAGS.

| Metric | Value |
|---|---|
| Score correlation (MV vs PEC) | 1.000 |
| Maximum score change | 0.044 |
| Agents with rank changes | 0 / 2 |
| Average item ambiguity | 0.047 |
| Ambiguous items ($>0.3$) | 51 (5.31%) |

The following tables present evaluation metrics for MSLR, a medical systematic literature review dataset with 6 LLM agents evaluated by 2 expert annotators: dataset summary statistics, agent score comparison across aggregation methods, ranking stability analysis, and annotator quality.

*Table 24.* MSLR dataset summary statistics.

| Metric | Value |
|---|---|
| Total annotations | 78 |
| Unique responses | 39 |
| Unique agents | 6 |
| Unique annotators | 2 |
| Avg. annotations per response | 2.00 |
| Unanimous agreement | 69.2% |
| Pairwise agreement | 69.2% |

*Table 25.* Agent scores on MSLR across scoring methods.

| Agent | MV | DS | PEC | $\Delta$(PEC-MV) | Items |
|---|---|---|---|---|---|
| VNCH8M | 0.700 | 0.900 | 0.848 | +0.148 | 5 |
| SPNXTA | 0.750 | 0.833 | 0.787 | +0.037 | 6 |
| PX7SGV | 0.688 | 0.688 | 0.691 | +0.003 | 8 |
| JB6Z8F | 0.562 | 0.688 | 0.653 | +0.091 | 8 |
| AQ85CE | 0.625 | 0.625 | 0.625 | +0.000 | 4 |
| 8FWF5T | 0.562 | 0.562 | 0.547 | $-0.015$ | 8 |

*Table 26.* Ranking stability on MSLR (lower is more stable).

| Method | Mean Rank Std | Mean Rank Range |
|---|---|---|
| Majority Vote | 0.503 | 1.000 |
| Dawid–Skene (Hard) | 0.503 | 1.000 |
| Posterior Expected Credit | 0.587 | 1.167 |

*Table 27.* Annotator quality on MSLR (both annotators).

| Annotator | Accuracy | Leniency | Strictness |
|---|---|---|---|
| annotator_0 | 0.794 | 0.253 | 0.219 |
| annotator_1 | 0.652 | 0.292 | 0.145 |

*Table 28.* Most ambiguous responses on MSLR.

| Response ID | Ambiguity | Confidence | Pred. |
|---|---|---|---|
| mslr_Cochrane_CD001487_SPNXTA | 0.261 | 0.739 | 1 |
| mslr_Cochrane_CD002873_8FWF5T | 0.261 | 0.739 | 1 |
| mslr_Cochrane_CD008510_JB6Z8F | 0.257 | 0.743 | 2 |
| mslr_Cochrane_CD006883_VNCH8M | 0.257 | 0.743 | 2 |
| mslr_Cochrane_CD008493_SPNXTA | 0.257 | 0.743 | 2 |
| mslr_Cochrane_CD008493_JB6Z8F | 0.257 | 0.743 | 2 |
| mslr_Cochrane_CD003700_VNCH8M | 0.257 | 0.743 | 2 |
| mslr_Cochrane_CD003700_AQ85CE | 0.132 | 0.868 | 1 |
| mslr_Cochrane_CD003700_JB6Z8F | 0.132 | 0.868 | 1 |
| mslr_Cochrane_CD003700_PX7SGV | 0.132 | 0.868 | 1 |

*Table 29.* Key findings on MSLR.

| Metric | Value |
|---|---|
| Score correlation (MV vs PEC) | 0.824 |
| Maximum score change | 0.148 |
| Agents with rank changes | 5 / 6 |
| Average item ambiguity | 0.103 |
| Highly ambiguous items ($>0.3$) | 0 (0.00%) |

The following figures present Mean Squared Error (MSE) and Ranking Accuracy across synthetic ablation configurations. For each ablation condition, we show both MSE with 95% confidence intervals and ranking accuracy comparing Majority Vote, Dawid–Skene, and PEC

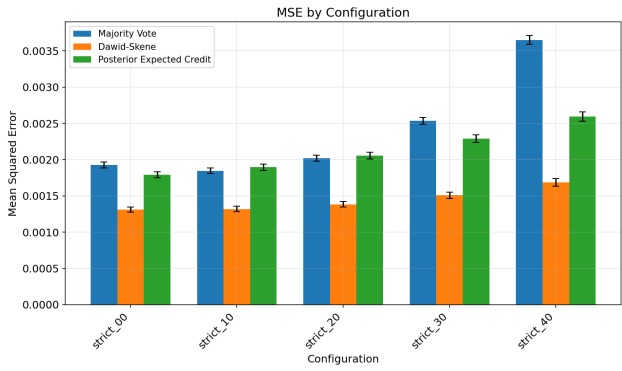

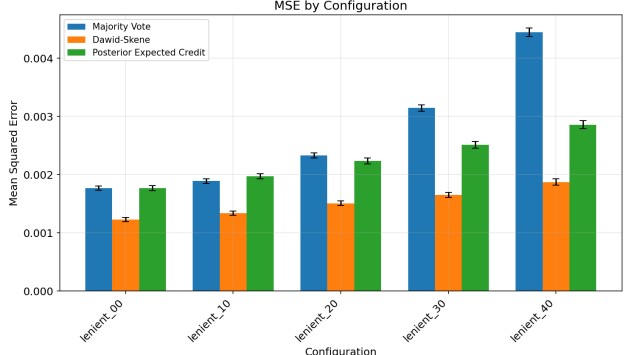

*Figure 4.* MSE across varying proportions of biased annotators. **Top:** Strict annotators (0–40%). **Bottom:** Lenient annotators (0–40%). Dawid–Skene achieves the lowest error across all configurations.

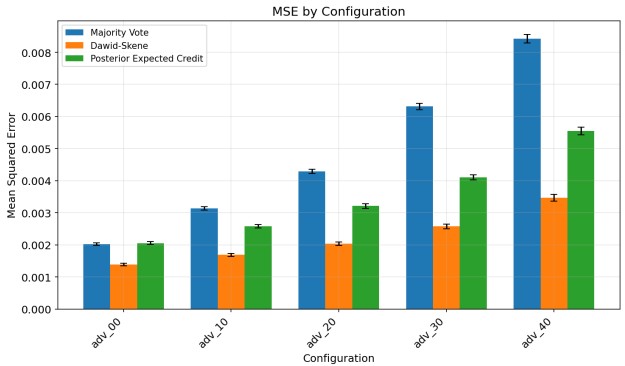

*Figure 3.* Robustness of Disagreement-Aware Aggregation Methods to Adversarial Annotators. MSE across different annotation aggregation methods as the fraction of adversarial annotators increases from 0% to 40%. Posterior Expected Credit and Dawid–Skene substantially outperform Majority Vote, with gaps widening at higher adversarial fractions.

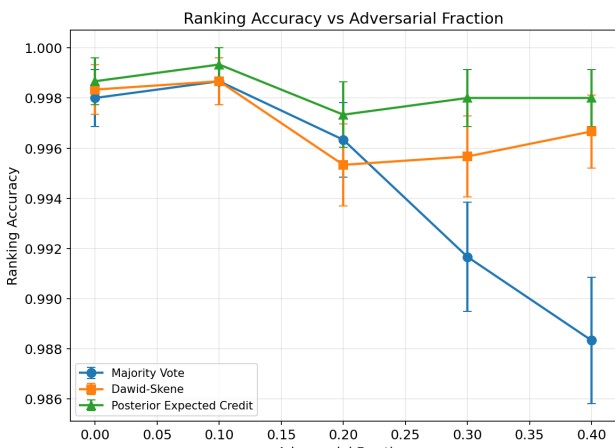

*Figure 5.* Ranking Accuracy vs Adversarial Fraction. Ranking accuracy (with 95% confidence intervals) as the fraction of adversarial annotators increases from 0% to 40%. Posterior Expected Credit maintains near-perfect accuracy across all fractions. Majority Vote drops from 0.998 to 0.988 at 40% adversarial fraction.

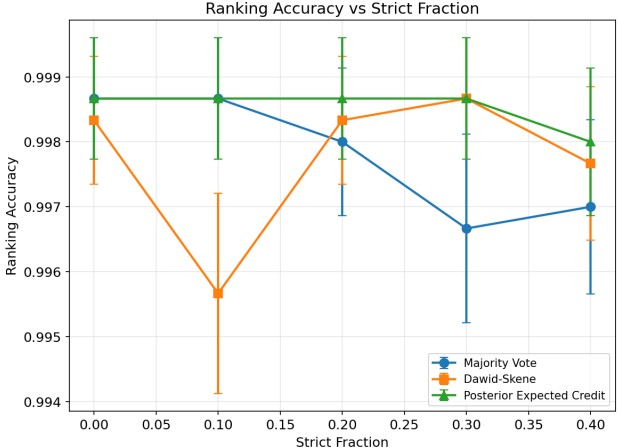

*Figure 6.* Ranking Accuracy Across Varying Proportions of Strict Annotators. Ranking accuracy (with 95% confidence intervals) as the fraction of strict annotators increases from 0% to 40%. Posterior Expected Credit maintains stable performance. Majority Vote degrades and Dawid–Skene exhibits volatility with a performance dip at 10% strict fraction.

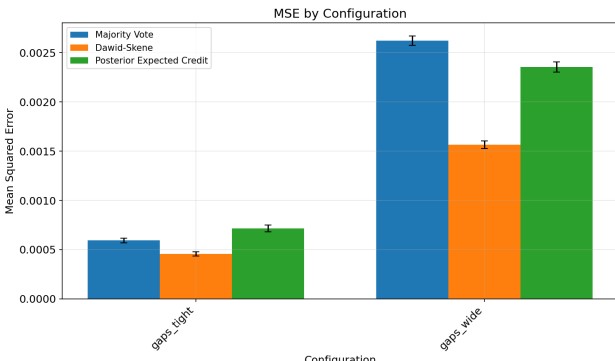

*Figure 8.* MSE Across Agent Quality Configurations. MSE (with 95% confidence intervals) comparing aggregation methods under tight and wide quality gaps among agents. The tight configuration uses agent qualities [0.85, 0.80, 0.70, 0.55, 0.35, 0.20]; the wide configuration uses [0.75, 0.70, 0.65, 0.60, 0.55, 0.50]. Dawid–Skene achieves the lowest error in the tight configuration (0.00047). Majority Vote error increases substantially in the wide configuration (0.00265), while Dawid–Skene (0.00159) and Posterior Expected Credit show smaller increases.

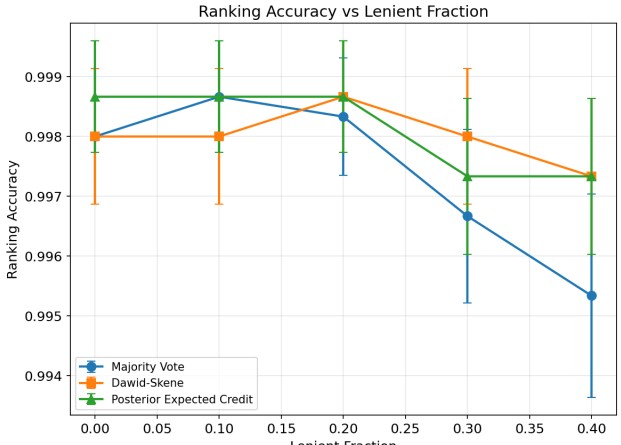

*Figure 7.* Ranking Accuracy Across Varying Proportions of Lenient Annotators. Ranking accuracy (with 95% confidence intervals) as the fraction of lenient annotators increases from 0% to 40%. Posterior Expected Credit maintains stable performance around 0.9975. Majority Vote degrades from 0.9982 to 0.9952 at 40% lenient fraction. Dawid–Skene remains relatively stable until 30% lenient fraction, then declines slightly.

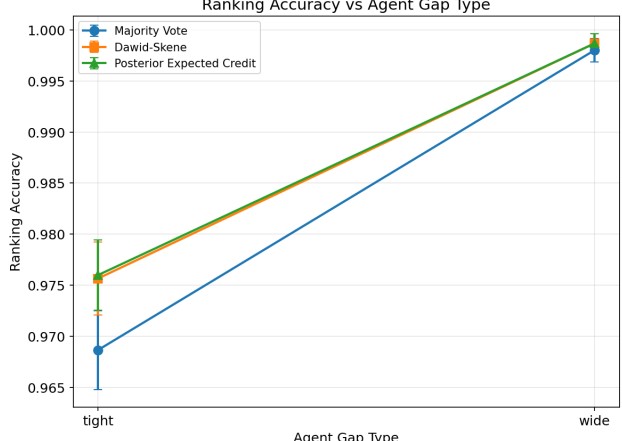

*Figure 9.* Ranking Accuracy vs Agent Gap Type. Ranking accuracy (with 95% confidence intervals) comparing aggregation methods under tight and wide quality gaps among agents. Dawid–Skene and Posterior Expected Credit converge near 1.000 in the wide configuration. Majority Vote increases from 0.9684 in the tight configuration to 0.9982 in the wide configuration, trailing the other methods by approximately 0.003 in the tight configuration.

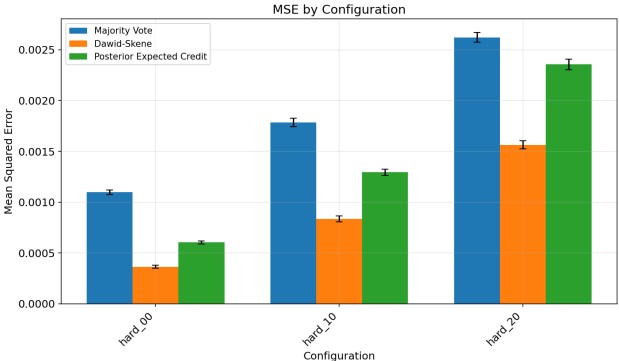

*Figure 10.* MSE Across Varying Percentages of Hard Items. MSE (with 95% confidence intervals) comparing aggregation methods as the percentage of hard items increases from 0% to 20%. Dawid–Skene achieves the lowest error across all configurations. Majority Vote error increases from 0.00110 at 0% hard items to 0.00265 at 20% hard items. Posterior Expected Credit error increases from 0.00058 to 0.00231 across the same range.

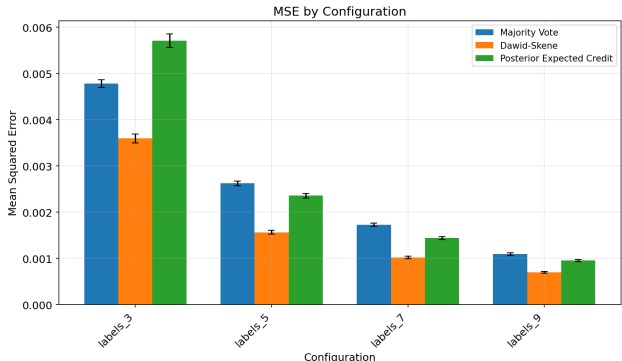

*Figure 12.* MSE Across Varying Numbers of Labels Per Item. MSE (with 95% confidence intervals) comparing aggregation methods as the number of labels per item increases from 3 to 9. Dawid–Skene achieves the lowest error across all configurations. Majority Vote error decreases from 0.00478 with 3 labels to 0.00110 with 9 labels. Posterior Expected Credit error decreases from 0.00560 to 0.00098 across the same range.

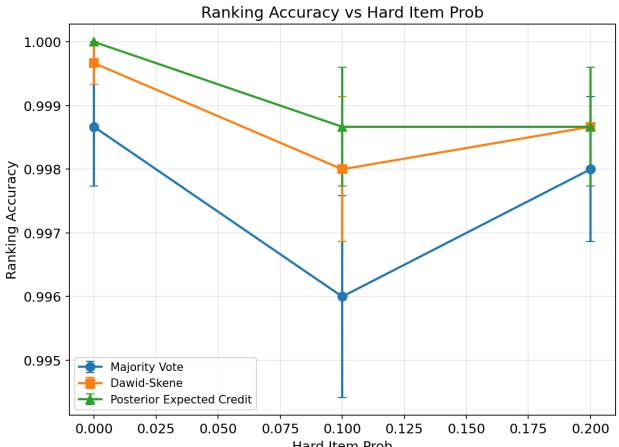

*Figure 11.* Ranking Accuracy vs Hard Item Probability. Ranking accuracy (with 95% confidence intervals) as the probability of hard items increases from 0.0 to 0.20. Posterior Expected Credit remains near 1.000 across all probabilities. Majority Vote declines from 0.9987 at 0.0 probability to a minimum of 0.9960 at 0.10 probability, then recovers to 0.9980 at 0.20 probability. Dawid–Skene declines monotonically from 0.9995 to 0.9985.

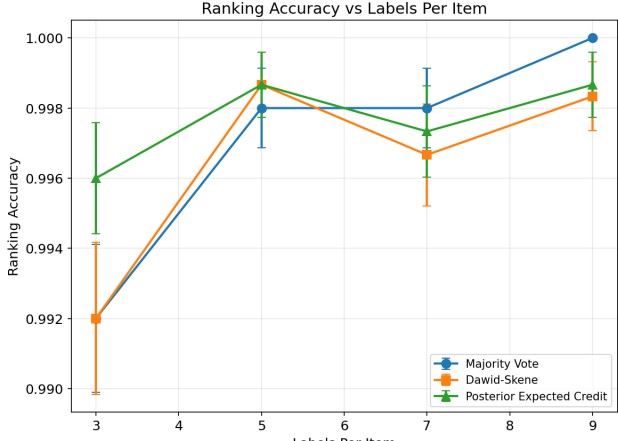

*Figure 13.* Ranking Accuracy vs Labels Per Item. Ranking accuracy (with 95% confidence intervals) comparing aggregation methods as the number of labels per item increases from 3 to 9. Majority Vote improves monotonically from 0.9948 with 3 labels to 1.0000 with 9 labels. Dawid–Skene and Posterior Expected Credit show non-monotonic behavior, peaking near 5 labels before declining slightly, then recovering at 9 labels. All three methods converge near 1.000 at 9 labels per item.

The following figures visualize agent score comparisons across three aggregation methods (Majority Vote, Dawid–Skene Hard, and PEC) for MT-Bench, ConvAbuse, QAGS, and MSLR datasets.

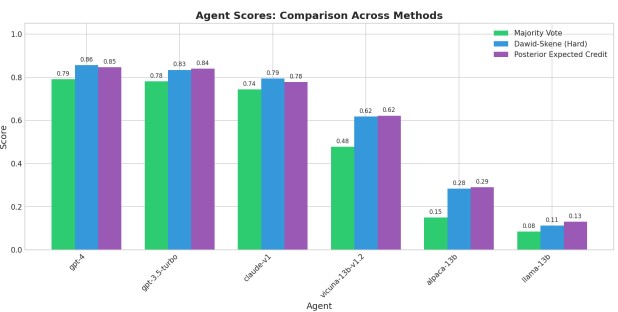

*Figure 14.* Agent scores comparison across three evaluation methods on MT-Bench for six agents: alpaca-13b, claude-v1, gpt-3.5-turbo, gpt-4, llama-13b, and vicuna-13b-v1.2. Scores shown for Majority Vote (green), Dawid–Skene Hard (blue), and Posterior Expected Credit (purple).

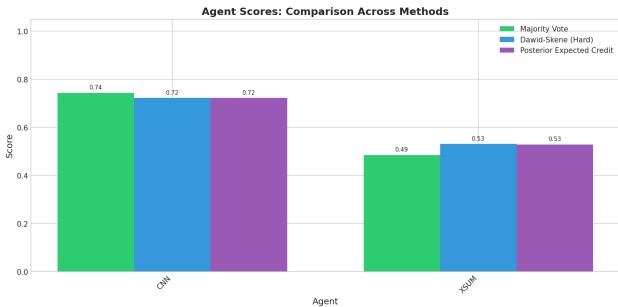

*Figure 16.* Agent scores comparison across methods on QAGS for two summarization models (CNN, XSUM). Scores shown for Majority Vote (green), Dawid–Skene Hard (blue), and Posterior Expected Credit (purple).

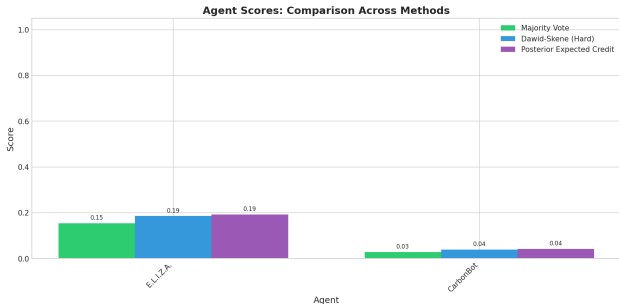

*Figure 15.* Agent scores comparison across three evaluation methods on ConvAbuse for two agents: E.L.I.Z.A. and CarbonBot. Scores shown for Majority Vote (green), Dawid–Skene Hard (blue), and Posterior Expected Credit (purple).

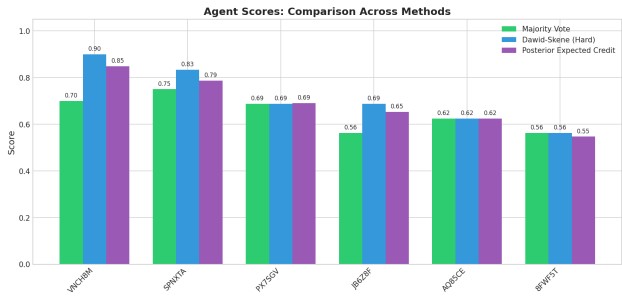

*Figure 17.* Agent scores comparison across three evaluation methods on MSLR for six agents: PX7SGV, 8FWF5T, SPNXTA, AQ85CE, VNCH8M, and JB6Z8F. Scores shown for Majority Vote (green), Dawid–Skene Hard (blue), and Posterior Expected Credit (purple)

