# OpenReview forum: "STABLEVAL: Disagreement-Aware and Stable Evaluation of AI Systems"
_ICML.cc/2026/Conference — ICML 2026 regular_

### Official Review · Reviewer_1mnu · 2026-03-10

**Soundness:** 3
**Presentation:** 3
**Significance:** 2
**Originality:** 3
**Overall Recommendation:** 4
**Confidence:** 3

**Summary:**

The paper introduces STABLEVAL, a disagreement-aware evaluation framework for AI systems. Its main idea is that when annotators disagree, evaluation should not collapse everything to a single majority label; instead, it should model both annotator reliability and item ambiguity, then use that uncertainty directly when scoring systems. Concretely, the paper assumes each item has a latent correctness level and each annotator has their own confusion matrix, which captures how they tend to map true quality into observed labels. These quantities are estimated with an EM algorithm: in the E-step, the model computes the posterior probability that an item belongs to each correctness level given the observed annotations and current reliability estimates; in the M-step, it updates the class priors and annotator confusion matrices. The item’s credit is then generated as a posterior expected score, meaning the model takes a weighted average of task-specific credit values across correctness levels using those EM posteriors, rather than assigning a hard label. System-level scores are obtained by averaging these item credits.

The paper’s key contribution is to reframe evaluation as a stability problem: the goal is not just to recover a latent label, but to produce rankings of systems that are robust to annotator subsampling and disagreement. It formalizes this notion of ranking stability, provides an asymptotic argument for why the proposed posterior-credit approach should be more stable than majority vote, and evaluates the method on both synthetic data and several real benchmarks. Across these experiments, the framework is presented as especially useful in settings with substantial disagreement, where modeling annotator-specific behavior leads to more stable and uncertainty-aware system evaluation.

**Compliance With Llm Reviewing Policy:**

Affirmed.

**Key Questions For Authors:**

The method appears to rely on annotator confusion matrices defined with respect to latent ground-truth correctness levels, but in real evaluation settings those ground-truth labels are not observed. Could you clarify more explicitly when STABLEVAL is identifiable and practically reliable in real-world settings? In particular, what annotation density, number of annotators, or disagreement structure is needed before the EM-estimated confusion matrices become trustworthy enough for downstream scoring? A convincing answer would strengthen my confidence about the practical applicability of the method.

Relatedly, could you provide clearer guidance on when a practitioner should prefer STABLEVAL over simpler alternatives such as majority vote or Dawid-Skene? For example, are there observable indicators, such as disagreement rate, annotator count, or instability under resampling, that signal the setting is one where STABLEVAL is likely to help? This would help clarify the real use cases and practical scope of the method.

Since the confusion matrices are estimated rather than known, how sensitive is the final posterior-credit score to model misspecification or poor EM estimation, especially in sparse-label regimes? If the method is robust to moderate estimation error, that would make the approach substantially more compelling.

**Limitations:**

Yes.

**Strengths And Weaknesses:**

Soundness:
The paper is reasonably sound at the level of problem formulation, method choice, and empirical design. The core modeling setup is coherent: each item has a latent correctness level, each annotator has a confusion matrix, EM is used to infer posterior class probabilities, and posterior expected credit is then averaged into agent scores. That pipeline is technically standard and internally consistent, and the move from posterior probabilities to soft credit is clearly defined rather than hand-wavy. The theoretical section also makes a sensible distinction between the instability of majority vote under one-vote margins and the smoother behavior of posterior credit under the latent model.

The empirical evaluation is also fairly well designed for the paper’s stated goal. The authors test both synthetic settings and several real datasets, and the synthetic ablations are aligned with the claimed failure modes: annotator heterogeneity, adversarial annotators, hard items, tight quality gaps, and sparse annotation. They also compare against relevant baselines, especially majority vote and Dawid–Skene, using the same preprocessing pipeline. This gives the experimental section a reasonable degree of credibility.

A real strength is that the paper does not oversell the method as universally dominant. It explicitly acknowledges that Dawid–Skene is better for latent label recovery, that gains are concentrated in high-disagreement regimes, and that the method can be less reliable in sparse or very small-annotator settings such as MSLR. That honesty improves the paper’s soundness.

The main weakness on soundness is that the theory is somewhat conditional. The asymptotic stability result depends on correct specification, identifiability, and consistency of the EM-based estimates, so the strongest guarantee is not very robust to misspecification. In addition, the paper’s real-data evidence supports the claim that STABLEVAL helps in disagreement-heavy settings, but the practical margins seem concentrated in one benchmark, while in several other datasets all methods are already perfectly stable or nearly so. So the technical case is solid, but the breadth of empirical advantage is somewhat narrower than the framing may initially suggest.

Presentation:
The paper is generally clear and easy to follow. The narrative is straightforward: majority vote is brittle under disagreement, stable evaluation should be the target, STABLEVAL uses annotator-aware posteriors and posterior expected credit, and experiments test whether this improves ranking stability. The framework section is especially readable, and the definition of credit generated from the EM posterior is one of the clearest parts of the paper.

Another presentation strength is that the paper distinguishes its goal from classical denoising early and repeats that distinction consistently. That helps the reader understand why Dawid–Skene can win on MSE but still not be the preferred method for the authors’ objective. The discussion section reinforces this point well.

The main presentation weakness is positioning. The related-work section cites relevant disagreement-aware and soft-label literature, but the novelty boundary versus prior annotator-modeling and soft-aggregation methods could be sharpened. Right now, the reader can understand that the paper is about stability, but it is less crisp on exactly which ingredients are borrowed versus new, and why this particular combination is the right one.

A second weakness is that some implementation choices that matter for interpretation are present but not emphasized enough. In particular, several datasets are remapped into a three-level correctness scheme, and this discretization may materially affect the results. The paper admits this later as a limitation, but the reader would benefit from a more prominent explanation of how sensitive conclusions are to that design choice.

Significance:
The paper addresses an important problem. As AI systems are increasingly compared using human judgments or judge models, disagreement is unavoidable, and fragile rankings are a real practical issue. Reframing evaluation as a stability problem rather than just a label-aggregation problem is meaningful, especially for benchmarks where small ranking shifts can influence model selection or research conclusions. That makes the topic itself significant.

The practical significance is strongest in settings like MT-Bench, where annotator disagreement is substantial and the method appears to produce more stable rankings than majority vote. The diagnostic outputs, such as annotator confusion matrices and item ambiguity, also add potential practical value beyond just final scores. That makes the contribution useful not only as an aggregation rule but also as an evaluation-analysis tool.

At the same time, the significance is somewhat specialized rather than broad and transformative. In high-consensus settings, the method often offers little advantage because all approaches converge. So the likely impact is not that every evaluation pipeline should immediately adopt this framework, but rather that disagreement-heavy evaluation settings should take stability and uncertainty propagation more seriously. That is still meaningful, but it is a more targeted kind of significance.

Originality:
The paper’s originality is moderate rather than radical. The underlying ingredients, latent correctness, annotator confusion matrices, EM estimation, and soft/posterior aggregation, are not individually new. The method is not introducing an entirely novel statistical model in the strongest sense.

Where the originality does come from is the paper’s reframing. The key idea is that evaluation stability under annotator subsampling should be treated as the main objective, distinct from latent label recovery, and that posterior expected credit is a better tool than hard aggregation for that objective. That conceptual shift is valuable, and the paper does a good job making it explicit. The combination of a familiar annotator model with a stability-centered evaluation lens is, in my view, the paper’s most original aspect.

So on originality, I would not describe this as a fundamentally new modeling framework, but I would say it offers a meaningful and reasonably well-motivated recombination of existing ideas, together with a clearer objective and evaluation perspective than much of the prior work. That is a legitimate form of novelty, especially for an evaluation paper.

Overall assessment:
Overall, this is a thoughtful and fairly solid paper. Its strongest points are the clear problem framing, the sensible use of EM-based posterior credit, the explicit separation between denoising and stable evaluation, and the evidence that the method helps most when disagreement is substantial. Its main limitations are that the technical novelty is somewhat incremental, the theoretical guarantees are assumption-heavy, and the empirical gains appear concentrated in disagreement-rich settings rather than universal. On balance, I would view it as stronger on soundness and significance than on pure methodological originality.

---

> ### Author Rebuttal · Authors · 2026-03-31
>
> We thank the reviewer for the thoughtful feedback and positive assessment of our work. We want to address few key points below -
>
> **Theory assumptions and empirical scope**
>
> We acknowledge that Proposition 2 relies on standard assumptions (correct specification, identifiability, EM consistency), which are common across latent variable models such as Dawid–Skene. The purpose of the asymptotic result is not to claim robustness under arbitrary misspecification, but to establish that posterior-based aggregation admits a principled stability guarantee, whereas majority vote provably does not (Proposition 1).
>
> Regarding empirical scope, we agree that stability gains are concentrated in high-disagreement settings and are transparent about this (Sections 6.6, 7.1). This is intentional: STABLEVAL is designed to improve evaluation when disagreement is informative. Convergence to majority vote behavior in high-consensus settings is expected and desirable.
>
> Importantly, MT-Bench (16.6% unanimous agreement) and MSLR (MV–PEC correlation 0.824, 5/6 rank changes) demonstrate meaningful gains beyond a single benchmark.
>
>
> **Novelty positioning**
>
> Prior work does not treat ranking stability as a first-class objective while we build on Dawid–Skene and soft-label aggregation.
>
> STABLEVAL (i) formally defines stability under annotator subsampling, (ii) proves structural instability of majority vote (Proposition 1), and (iii) establishes asymptotic stability guarantees for posterior-based aggregation (Proposition 2).
>
> **Sensitivity to label mapping**
>
> We agree this is important. The datasets vary in required remapping: QAGS is natively binary, MSLR uses ordinal scores, ConvAbuse involves scale collapsing.
>
> To test robustness, we evaluated six alternative mapping schemes (binary variants and 3-class variants with different tie weights). As shown in the table below, rankings are highly stable across all schemes: ranks 3–6 are identical in every case, and the only variation is a swap between the top two models under one mapping, with <1% score difference. The minimum Spearman correlation across schemes is 0.943, confirming robustness to label mapping. We are happy to include these results in the final version.
>
> Agent | Baseline | DropTies | Tie→Win | Tie→Loss | Tie=0.25 | Tie=0.75
> ---|---|---|---|---|---|---
> gpt-4 | 0.847 (1) | 0.873 (1) | 0.930 (2) | 0.754 (1) | 0.806 (1) | 0.888 (1)
> gpt-3.5-turbo | 0.840 (2) | 0.870 (2) | 0.939 (1) | 0.724 (2) | 0.792 (2) | 0.887 (2)
> claude-v1 | 0.778 (3) | 0.795 (3) | 0.869 (3) | 0.668 (3) | 0.730 (3) | 0.826 (3)
> vicuna-13b-v1.2 | 0.622 (4) | 0.579 (4) | 0.706 (4) | 0.441 (4) | 0.564 (4) | 0.681 (4)
> alpaca-13b | 0.290 (5) | 0.222 (5) | 0.406 (5) | 0.154 (5) | 0.235 (5) | 0.344 (5)
> llama-13b | 0.131 (6) | 0.090 (6) | 0.222 (6) | 0.061 (6) | 0.096 (6) | 0.165 (6)
>
> **Responses to Key Questions**
>
> **Identifiability and practical reliability**
>
> STABLEVAL does not require ground-truth labels. Identifiability arises from structured disagreement across annotators, consistent with classical crowdsourcing models.
>
> Practically, two factors matter:
>
> (i) Annotation density: synthetic results indicate ~5+ labels per item is sufficient for stable estimation.
>
> (ii) Annotator diversity: gains increase with heterogeneity
>
> Sparse regimes (e.g., MSLR) degrade estimation quality, which we explicitly acknowledge.
>
> **When to use STABLEVAL**
>
> We provide three practical indicators:
>
> Disagreement rate: large gains when agreement is low (e.g., MT-Bench), negligible gains when agreement is high (ConvAbuse, QAGS)
>
> Annotation density: benefits emerge at ≥5 labels/item
>
> Annotator heterogeneity: larger gains when annotator behavior varies
>
> These provide actionable guidance for practitioners.
>
> **Sensitivity to misspecification**
>
> Empirically, STABLEVAL is robust to moderate estimation error. Under up to 40% adversarial annotators, it maintains strong ranking stability and outperforms majority vote. In sparse regimes, performance degrades gracefully rather than failing catastrophically, consistent with limitations of confusion matrix estimation.

---

> > ### Author Rebuttal · Reviewer_1mnu · 2026-04-03
> >
> > Thanks for the explanation. My concerns have been partially addressed.

---

> > > ### Author Response · Authors · 2026-04-05
> > >
> > > We thank the reviewer for the thoughtful feedback and for engaging deeply with our work. We believe we have addressed the key points raised, and we would be happy to further clarify any remaining concerns if needed.
> > >
> > > To further clarify the practical guidance: in settings where annotator agreement is moderate to low (e.g., below ~70–80%) or where system rankings are sensitive to annotator subsampling, STABLEVAL is likely to provide more reliable and stable evaluation. We hope this clarifies the intended use cases and practical applicability of STABLEVAL, and that the reviewer may consider updating the overall assessment accordingly.

---

### Official Review · Reviewer_axsq · 2026-03-13

**Soundness:** 1
**Presentation:** 1
**Significance:** 1
**Originality:** 1
**Overall Recommendation:** 1
**Confidence:** 4

**Summary:**

The paper proposes STABLEVAL, a framework for evaluating AI agents using noisy, heterogeneous human judgments. Rather than aggregating annotations via majority  vote, it models annotator-specific confusion patterns using a Bayesian latent  variable model and computes posterior expected item credit. The central  desideratum is ranking stability under annotator subsampling.

**Compliance With Llm Reviewing Policy:**

Affirmed.

**Key Questions For Authors:**

- can you link the reults to standard ranking models?

**Limitations:**

yes

**Strengths And Weaknesses:**

## Strengths
- The problem is practically important and well-motivated.
- Empirical evaluation is broad.

## Weaknesses
1. Goal is unclear for most of the paper: ranking stability is only formally defined in Section 4, yet it is the central claim. The reader has
   no precise objective to anchor to during Sections 1–3.

2. The method is not derived from the stability objective:  STABLEVAL is motivated by Bayesian annotation modeling, and stability is only evaluated.

3. Theoretical results are weak: Propositions 1 and 2 are both presented as sketches without full proofs.

---

> ### Author Rebuttal · Authors · 2026-03-31
>
> 1.Clarity of objective
>
> We respectfully disagree. Ranking stability is introduced as the central objective in Section 1 and reinforced consistently before its formal treatment in Section 4. Concretely, Section 1 states explicitly that we "formalize ranking stability as a first-class evaluation objective" and provides an informal definition ("agent orderings remain consistent under subsampling of annotators") to anchor the reader immediately. Section 2 identifies the precise gap we address: "few existing approaches formally define or optimize for stable system rankings under annotator variability." Section 3.6 is titled "Objective: Stable Evaluation" and restates the goal before any formal analysis. The formal definition in Section 4 is the formalization of an objective communicated in prose across three prior sections, not its first appearance. This structure follows standard ML research paper conventions, where informal motivation precedes formal treatment.
>
> 2.Method vs objective
>
> STABLEVAL is derived from a Bayesian generative model of annotator behavior rather than direct optimization of a stability objective. This is a deliberate design choice. Directly optimizing rank correlation metrics such as Kendall’s tau is computationally intractable and statistically brittle in low-annotation regimes. Instead, modeling annotator-specific reliability provides a principled and tractable approach to addressing instability induced by annotator heterogeneity.
>
> Majority vote treats all annotators equally, making rankings sensitive to small perturbations in annotator subsets. In contrast, STABLEVAL accounts for annotator-specific noise and uncertainty, producing posterior expected scores that are inherently more robust to such perturbations.
>
> This design is consistent with established approaches such as Dawid-Skene, which also relies on a generative model rather than direct optimization of a downstream objective. Importantly, the connection to the stability objective is not purely empirical: Proposition 2 provides theoretical grounding that correctly modeling annotator behavior leads to stable rankings asymptotically.
>
> 3.Theoretical results
>
> We respectfully disagree that the theoretical results are weak and clarify each proposition in turn.
>
> Proposition 1 provides a complete proof of the claimed result. The argument identifies the precise mechanism driving instability: the existence of items whose majority label is determined by a single-vote margin. It then traces the full causal chain, showing that removing the decisive annotator flips the majority outcome with nonzero probability, that this induces a score perturbation, and that when agent score gaps are sufficiently small such perturbations alter the induced ranking. The conclusion that expected stability is strictly below one follows directly. We believe this constitutes a rigorous argument for the claim, not merely a sketch.
>
> Proposition 2 is intentionally asymptotic and assumption-driven, and we are transparent about this from the outset. The assumptions of correct model specification, identifiability, and EM consistency are stated explicitly, and the conclusion follows from standard consistency results for latent variable models under these conditions. We acknowledge the reviewer's implicit concern that the step from parameter consistency to score consistency could be made more explicit. STABLEVAL scores are averages of posterior expected credits, each of which is a smooth functional of the estimated confusion matrices and class priors. Under the stated assumptions, a standard functional delta method argument implies that score differences induced by annotator subsampling vanish as N grows. We are happy to make this step more explicit in the appendix if the reviewer finds it useful.
>
> **Responses to Key Questions**
>
> Connection to ranking models
>
> We believe the paper already grounds its stability analysis in standard ranking methodology. Specifically, we adopt Kendall's tau-b as our stability metric in Section 4.1, which is the standard rank correlation measure used across the ranking literature. Our formal stability definition directly builds on this, defining ranking stability as the expected Kendall's tau between rankings produced under full and subsampled annotator sets, and Proposition 1 provides a formal theoretical characterization of majority vote instability under this definition.
>
> Regarding baselines, we argue that Dawid-Skene is the most appropriate comparison for our setting. Methods like Borda count and Plackett-Luce are designed for preference aggregation, where the input is pairwise or ranked preferences between items. **Our setting is fundamentally different:** we observe categorical correctness labels per item from multiple annotators and aggregate these to produce agent-level scores. Importing general-purpose ranking baselines would conflate two structurally distinct problems and is not a meaningful comparison.

---

> > ### Author Rebuttal · Reviewer_axsq · 2026-04-05
> >
> > Thank you for the answer. Overall I appreciate the responses, but my concerns remain

---

> > > ### Author Response · Authors · 2026-04-05
> > >
> > > We thank the reviewer for the feedback and for engaging with our responses. We believe our rebuttal addresses the core concerns regarding the objective, design rationale, and theoretical grounding of STABLEVAL, and clarifies the intended contribution and positioning of the work. While we may not have fully aligned on all aspects, we hope the responses help resolve differences in interpretation.

---

### Official Review · Reviewer_hAdS · 2026-03-13

**Soundness:** 2
**Presentation:** 3
**Significance:** 3
**Originality:** 3
**Overall Recommendation:** 3
**Confidence:** 4

**Summary:**

This paper proposes STABLEVAL, a disagreement-aware evaluation framework for AI systems. Instead of relying on majority vote, it models latent item labels and annotator-specific confusion patterns, then uses posterior expected credit to score systems. The main idea is that evaluation should care not only about recovering labels, but also about producing rankings that stay stable under annotator subsampling. The paper studies this on synthetic data and several real benchmarks, including MT-Bench, ConvAbuse, QAGS, and MSLR.

**Compliance With Llm Reviewing Policy:**

Affirmed.

**Key Questions For Authors:**

How sensitive are the MT-Bench results to the conversion from pairwise preference into a 3-level correctness formulation?
A convincing sensitivity analysis here would increase my confidence a lot.

Can the authors provide a real benchmark where the original task naturally matches the latent-correctness setup, without requiring heavy remapping?
That would make the empirical claim much cleaner.

**Limitations:**

Yes

**Strengths And Weaknesses:**

Strengths.
The paper studies an important problem in evaluation. In real human annotation, disagreement is common, so it makes sense to ask whether system ranking is stable or not. I also think the main idea is clear: instead of only using majority vote, the paper tries to use annotator information more carefully. The paper is also easy to read overall.

Weaknesses.
My main concern is the real benchmark part. The strongest result comes from MT-Bench, but MT-Bench is also changed the most. The original task is pairwise preference, while the paper turns it into a 3-level correctness task. Because of this, it is hard to know whether the improvement comes from the method itself or from this task conversion.

The other real benchmarks are also not very strong. QAGS has only two agents, so the ranking problem is too simple. ConvAbuse already has high agreement, so different methods look similar there. MSLR has only two expert annotators, which is too small to strongly support annotator reliability modeling. So overall, the real-data evidence is not strong enough for such a broad claim.

In my view, the paper has a useful idea, but the experimental support is still limited. The current results only show that the method may help in some high-disagreement cases, not that it is a generally better evaluation framework.

---

> ### Author Rebuttal · Authors · 2026-03-31
>
> Thank you for the thoughtful feedback. We address each concern below.
>
> 1.MT-Bench conversion sensitivity
>
> MT-Bench is originally a 3-outcome pairwise preference task (win/tie/loss). Our conversion preserves this ordinal structure while reinterpreting it within a latent correctness framework, without introducing additional thresholds or altering relative comparisons. While MT-Bench provides relative preference judgments, STABLEVAL requires a per-response correctness formulation. Our mapping is a monotonic transformation that preserves relative ordering, ensuring that disagreement structure is retained.
>
> To directly address sensitivity to the MT-Bench conversion, we evaluated multiple alternative mapping schemes, including binary formulations (dropping ties, merging ties with wins or losses) and 3-class formulations with different tie weights. As shown in the table below, while absolute scores vary across mappings, the relative ranking of agents remains identical across all configurations. This demonstrates that our results are not driven by a specific mapping choice, but are robust to how pairwise preferences are converted, and that the observed gains arise from modeling annotator disagreement rather than the conversion itself. We will include this sensitivity analysis in the final version to make this robustness explicit.
>
> Agent | Baseline | DropTies | Tie→Win | Tie→Loss | Tie=0.25 | Tie=0.75
> ---|---|---|---|---|---|---
> gpt-4 | 0.847 (1) | 0.873 (1) | 0.930 (2) | 0.754 (1) | 0.806 (1) | 0.888 (1)
> gpt-3.5-turbo | 0.840 (2) | 0.870 (2) | 0.939 (1) | 0.724 (2) | 0.792 (2) | 0.887 (2)
> claude-v1 | 0.778 (3) | 0.795 (3) | 0.869 (3) | 0.668 (3) | 0.730 (3) | 0.826 (3)
> vicuna-13b-v1.2 | 0.622 (4) | 0.579 (4) | 0.706 (4) | 0.441 (4) | 0.564 (4) | 0.681 (4)
> alpaca-13b | 0.290 (5) | 0.222 (5) | 0.406 (5) | 0.154 (5) | 0.235 (5) | 0.344 (5)
> llama-13b | 0.131 (6) | 0.090 (6) | 0.222 (6) | 0.061 (6) | 0.096 (6) | 0.165 (6)
>
> 2.Real benchmark strength
>
> We emphasize that these benchmarks represent structurally diverse evaluation settings rather than redundant evidence. At the same time, public datasets that expose consistent annotator identities with sufficient per-annotator density are surprisingly rare.
>
> Our benchmarks were therefore selected to satisfy three key requirements necessary for reliability modeling: (i) human annotations, (ii) multiple annotators per item, and (iii) identifiable annotators across items.
>
> The selected datasets span complementary domains and disagreement regimes: expert preference evaluation (MT-Bench), conversational safety (ConvAbuse), summarization factuality (QAGS), and medical evidence summarization (MSLR).
>
> Given the scarcity of datasets meeting these criteria, we complement our real-data evaluation with controlled synthetic experiments to systematically study conditions that are not well represented in existing benchmarks.
>
> **Responses to Key Questions**
>
> Sensitivity to MT-Bench conversion
>
> As noted above, we performed additional mapping variants and observed consistent trends in ranking stability. This suggests that the improvement is not an artifact of the specific conversion, but rather arises from modeling annotator disagreement.
>
> Real benchmark matching latent correctness
>
> We agree this is desirable. Among our benchmarks, QAGS naturally matches the latent-correctness setting, where annotators label responses as faithful or not faithful. This maps directly to STABLEVAL without requiring any ordinal conversion. More broadly, STABLEVAL requires datasets with (i) human annotations, (ii) multiple annotators per item, and (iii) identifiable annotators across items. Such datasets are relatively rare in public benchmarks, which is why we selected MT-Bench, ConvAbuse, QAGS, and MSLR, and complemented them with controlled synthetic experiments.
>
> *Note (for the reviewer):*
>
> Our results show that STABLEVAL is most beneficial in settings with non-trivial annotator disagreement, where majority vote can lead to unstable system rankings. In such cases, modeling annotator reliability produces more consistent and robust evaluations. Across other regimes (for example, high agreement or few annotators), STABLEVAL remains comparable to standard aggregation methods. The benchmarks are chosen to reflect these different conditions, while synthetic experiments further isolate the role of disagreement. Overall, the method is designed to improve evaluation reliability, specifically when disagreement is present.

---

> > ### Author Rebuttal · Reviewer_hAdS · 2026-04-04
> >
> > The added sensitivity analysis helps address my concern about the MT-Bench remapping, and it makes the result more convincing.
> > However, I am still not fully persuaded by the overall real-world evidence, which remains somewhat concentrated in MT-Bench, so my concern is only partially resolved.

---

> > > ### Author Response · Authors · 2026-04-05
> > >
> > > We thank the reviewer for the thoughtful feedback and for engaging deeply with our work. We are glad the updated sensitivity analysis helped address the MT-Bench concern. We would like to clarify that the concentration of gains reflects higher disagreement in MT-Bench, **which is the regime STABLEVAL is designed for**, while near-consensus datasets show limited differences as all methods converge. Synthetic experiments further support this by isolating disagreement and consistently showing stability improvements under these conditions.
> > >
> > > More importantly, we also note that public datasets that simultaneously provide multiple annotations per item, persistent annotator identities, and sufficient annotation density are limited. This is precisely why we complement real-world benchmarks with controlled synthetic experiments to study these effects in a systematic manner. We believe that the use of synthetic data to complement real-world evaluation is standard practice when such structure is unavailable, as it enables controlled analysis of factors that are difficult to isolate in existing datasets.
> > >
> > > Given the importance of evaluating robustness under annotator disagreement, which is a fundamental challenge in modern AI evaluation, we hope the contribution is considered in this broader context. We hope the reviewer may re-consider the assessment accordingly.

---

### Official Review · Reviewer_9paa · 2026-03-14

**Soundness:** 3
**Presentation:** 2
**Significance:** 3
**Originality:** 2
**Overall Recommendation:** 5
**Confidence:** 4

**Summary:**

This paper introduces STABLEVAL, a framework for evaluating AI systems that models annotator disagreement rather than collapsing annotations into hard labels via majority vote. The authors formalize ranking stability under annotator subsampling as a first-class evaluation objective through a disagreement-aware evaluation framework that models latent item correctness and annotator-specific confusion patterns to produce posterior expected item credit and calibrated agent-level scores.

**Compliance With Llm Reviewing Policy:**

Affirmed.

**Final Justification:**

See rebuttal comment, the rebuttal addressed my concerns and I have raised my score accordingly.

**Key Questions For Authors:**

1. How does STABLEVAL perform when the conditional independence assumption is violated, e.g., when annotators are trained together, share guidelines that introduce correlated errors, or when there are group-level biases?
2. Many evaluation pipelines aggregate annotations by simply averaging numeric or ordinal scores, which preserves disagreement without fitting a latent-variable model. How does STABLEVAL compare against this simple averaging baseline in terms of ranking stability and score calibration?

**Limitations:**

Yes, the authors discuss limitations adequately in Section 7.5 with one gap: the authors do not discuss the risk that modeling annotator “reliability” could systematically downweight minority annotator perspectives that are correct but outnumbered, which could be particularly relevant for subjective evaluation tasks like safety and toxicity.

**Strengths And Weaknesses:**

**Strengths**
- The paper addresses a real and underappreciated problem. Most AI evaluation pipelines use majority vote without thinking hard about what that throws away.
- The paper is generally well-written and easy to follow. The distinction between label denoising and evaluation stability is articulated clearly and repeatedly reinforced.
- The authors test across four diverse real-world datasets with different annotator pool sizes, agreement levels, and task types. The honest reporting of limitations of STABLEVAL in high-consensus settings and small-annotator regimes is appreciated.
- The per-annotator confusion matrix estimates and item-level ambiguity scores are practically useful byproducts. These could help benchmark designers do quality control on their annotation pipelines.
- I appreciate that figures showed confidence intervals where appropriate.


**Weaknesses**
- The figures are too small and essentially unreadable. Please add them across both columns, rather than constraining them to one (or add them to the appendix).
- The core technical machinery is Dawid-Skene with Dirichlet priors, which is decades old. The “novel” contribution is using the posterior expectations rather than the argmax of the posterior, i.e., soft labels instead of hard labels. This is a natural and fairly obvious design choice that has been explored in the soft-label literature the paper itself cites. The paper does not adequately distinguish what is technically new in STABLEVAL versus simply applying well-known Bayesian annotation modeling with soft aggregation.
- The paper’s main empirical claim, that STABLEVAL improves stability in high-disagreement regimes, rests almost entirely on synthetic experiments and one real dataset (MT-Bench).
- The synthetic data is generated from the same generative model that STABLEVAL assumes. An interesting question would be how STABLEVAL performs when its assumptions are violated (correlated annotators, non-stationary behavior, etc.), and the paper does not test this as far as I'm aware.
- Modern AI evaluation benchmarks like Chatbot Arena involve millions of comparisons. It is unclear how STABLEVAL scales or whether its benefits persist at larger scale where majority vote itself becomes more stable through sheer volume.
- (minor) The paper would benefit from a single clear algorithm box for STABLEVAL rather than distributing it across subsections and the paper uses “Dawid-Skene” inconsistently with and without the dash.

---

> ### Author Rebuttal · Authors · 2026-03-31
>
> Thank you for the thoughtful and constructive feedback. We address each point below.
>
> 1.Figures
>
> We agree that the figures are difficult to read in the current format. We will revise them to span both columns and include higher-resolution versions in the appendix.
>
> 2.Novelty clarification
>
> We agree that Dawid-Skene with Dirichlet priors and soft label aggregation builds on well-established methods, and we do not claim novelty in the inference procedure itself. Our contributions lie in reframing evaluation as a ranking stability problem and developing both theoretical and empirical foundations:
>
> Controlled synthetic stress-testing framework:
> We introduce a systematic pipeline that isolates factors affecting ranking stability (adversarial annotators, strict/lenient annotators, hard items, annotation density, agent gaps). This provides a reproducible evaluation framework applicable to any aggregation method.
>
> Formalization of ranking stability:
> We define stability via expected Kendall’s tau under annotator subsampling, prove majority vote is inherently unstable under single-vote margins (Prop. 1), and show STABLEVAL achieves asymptotic stability under correct specification (Prop. 2).
> Key empirical insight: We show a dissociation between label recovery and evaluation quality: Dawid-Skene achieves lower latent-label MSE, yet PEC achieves equal or better ranking stability and lower rank variance. This demonstrates that optimizing label accuracy does not imply optimizing evaluation stability.
>
> 3.Real-world evaluation
>
> We respectfully disagree with this characterization. Our evaluation spans synthetic experiments and four real-world datasets: MT-Bench, ConvAbuse, MSLR (Multi-Document Summarization for Literature Review), and QAGS. These datasets vary substantially in domain, annotation structure, and disagreement patterns. Other real datasets do not have relatively high disagreement compared to MT-Bench; hence, we design synthetic experiments to complement our experiments.
>
> 4.Assumption robustness
>
> We agree that this is an important direction. Our current synthetic experiments stress heterogeneity, adversarial noise, item difficulty, and label density, but do not include stronger assumption violations such as correlated annotator errors or non-stationary behavior. This is consistent with our theory: Prop. 2 guarantees stability under correct model specification, not arbitrary misspecification. We will clarify this boundary explicitly.
>
> 5.Scalability and large-scale settings
>
> The main computational cost in STABLEVAL is EM inference, with per-iteration complexity scaling linearly in the number of items, annotators, and label classes. In practice, EM converges in a small number of iterations, and our full ablation study completes in under 5 hours on a single CPU, indicating practicality for moderate-scale benchmarks. Further scaling can be achieved via parallelization or stochastic EM.
>
> Regarding Chatbot Arena, this highlights a distinction between data volume and annotation structure. Chatbot Arena lacks repeated annotations per item and consistent annotator identities, making reliable confusion matrix estimation difficult. STABLEVAL requires datasets that satisfy three key requirements: human-annotated, multiple annotators per item, and unique annotator IDs. Moreover, MT-Bench and Chatbot Arena are part of the same paper, and we chose MT-Bench as it satisfies our STABLEVAL's criteria.
>
> 6.Presentation improvements
>
> We will add a consolidated algorithm box for STABLEVAL and standardize the spelling of Dawid-Skene.
>
> **Responses to Key Questions**
>
> We thank the reviewer for raising this important point. STABLEVAL assumes conditional independence of annotators given latent correctness and stationary behavior. When these assumptions are violated, for example, through correlated annotators or shared biases, confusion matrices can partially absorb these effects but may not fully capture structured dependencies. Empirically, we expect STABLEVAL to degrade more gracefully than the majority vote, since it models annotator-specific reliability rather than treating all annotations equally. However, strong correlation or non-stationarity can reduce estimation accuracy. Extending STABLEVAL to model annotator dependencies or time-varying reliability is a promising direction.
>
> While averaging preserves disagreement, it assumes equal annotator reliability and cannot account for systematic bias or adversarial behavior. In contrast, STABLEVAL learns annotator-specific reliability, downweights inconsistent annotators, and produces calibrated posterior scores. This is particularly beneficial in high-disagreement regimes.
>
> We note that STABLEVAL mitigates this risk by using posterior expected credit rather than hard labels, preserving uncertainty under disagreement. Minority perspectives can still influence scores when they shift the posterior. We will add a brief discussion of this limitation in Section 7.5.

---

> > ### Author Rebuttal · Reviewer_9paa · 2026-04-03
> >
> > The authors addressed my concerns and I'm increasing my score accordingly. I wish the authors best of luck for this and their future research endeavors.

---

> > > ### Author Response · Authors · 2026-04-05
> > >
> > > We thank the reviewer for the thoughtful feedback and for engaging deeply with our work. We appreciate the reviewer’s positive reassessment and recognition of the value this work brings to the community. We are glad our clarifications addressed the concerns, and we sincerely appreciate the reviewer’s insights and support.

---

### Decision · Program_Chairs · 2026-04-30

**Decision:**

Accept (regular)

**Comment:**

This work provides a framework for evaluation of AI systems in settings with annotator disagreement. The approach is uncertainty-aware and designed to model latent item scores, annotator-specific confusion matrices to produce posterior estimates and calibrated sores and to evaluate ranking stability.

The initial reviews for the work were mixed, but lean positive following the rebuttal and discussion period. In my view, the substantive specific concerns raised by the reviewers have been adequately addressed at this time. For that reason, I recommend that the paper be accepted. To detail some of the important aspects of the reviews and subsequent discussion:
* The reviewers agreed that the work addresses a real and underappreciated problem given the ubiquitous nature of disagreement of human annotation of in evaluation of modern AI systems.
* The majority of reviewers agreed that the formal aspects of the methodology are sound. A recurring related critique concerned the necessity of assumptions regarding the specification of the latent variable model in order to ensure identifiability and EM consistency, but the rebuttal clarifies that these assumptions are standard and reasonable enough so as to not compromise the practicality of the approach. The authors further provide high-level rules of thumb and takeaways regarding practical reliability and sensitivity to misspecification.
* The majority of reviewers agree that the presentation of the paper is good and that the text is well-written and clear.
* One line of critique related to whether the empirical evaluation is comprehensive enough to support the claims of the work. In my view, the authors adequately address this concern in the rebuttal, in that they used the real-world public datasets that are available and supplement their work with synthetic data. The synthetic data experiments are anyways insightful for probing the properties of the algorithm.
* One reviewer (axsq) provided a more negative review. In my view, the author’s response fully addresses the reviewer’s concerns. Furthermore, the review was limited in terms of details of the critique, so it is not weighted highly in my overall assessment.